# Structural Insights into Bortezomib-Induced Activation of the Caseinolytic Chaperone-Protease System in *Mycobacterium tuberculosis*

Biao Zhou[1,2,3,4,13], Yamin Gao [4,5,13], Heyu Zhao[4,5,13], Banghui Liu[4], Han Zhang[4,6], Cuiting Fang [4,5], Hang Yuan[4,5], Jingjing Wang[4], Zimu Li[2,3,4], Yi Zhao[4,5], Xiaodong Huang[2,4], Xiyue Wang[7], A. Sofia. F. Oliveira [8], James Spencer [9], Adrian J. Mulholland [8], Steven G. Burston [10], Jinxing Hu[1], Ning Su[1], Xinwen Chen [3,11] ✉, Jun He [4,12] ✉, Tianyu Zhang [4] ✉ & Xiaoli Xiong [4] ✉

The caseinolytic protease (Clp) system has recently emerged as a promising anti-tuberculosis target. The anti-cancer drug bortezomib exhibits potent anti-mycobacterial activity and binds to *Mycobacterium tuberculosis* (*Mtb*) Clp protease complexes. We determine cryo-EM structures of *Mtb* ClpP1P2, ClpC1P1P2 and ClpXP1P2 complexes bound to bortezomib in different conformations. Structural and biochemical data indicate that sub-stoichiometric binding by bortezomib to the protease active sites orthosterically activates the *Mtb*ClpP1P2 complex. Bortezomib activation of *Mtb*ClpP1P2 induces structural changes promoting the recruitment of the chaperone-unfoldases, *Mtb*ClpC1 or *Mtb*ClpX, facilitating holoenzyme formation. The structures of the *Mtb*ClpC1P1P2 holoenzyme indicate that *Mtb*ClpC1 motion, induced by ATP rebinding at the *Mtb*ClpC1 spiral seam, translocates the substrate. In the *Mtb*ClpXP1P2 holoenzyme structure, we identify a specialized substrate channel gating mechanism involving the *Mtb*ClpX pore-2 loop and *Mtb*ClpP2 N-terminal domains. Our results provide insights into the intricate regulation of the *Mtb* Clp system and suggest that bortezomib can disrupt this regulation by sub-stoichiometric binding at the *Mtb* Clp protease sites.

Despite being a preventable and curable disease, tuberculosis (TB) is still one of the leading causes of death worldwide to be caused by a single infectious agent. Both the United Nations (UN) and World Health Organization (WHO) have made it a priority to end this global epidemic by 2030. Central to this is the development and mechanistic understanding of new anti-TB therapies as strains of *Mycobacterium tuberculosis* (*Mtb*) emerge that are resistant to even potent antibiotics, such as rifampicin[1].

One emerging target for anti-TB therapy development is the caseinolytic protease (Clp) system. This system is present in most eubacteria, as well as in the mitochondria and plastids of eukaryotes[2]. It plays a critical role in maintaining protein homeostasis, exercising protein quality control, and regulating various cellular processes by modulating the levels of key regulatory proteins[3], including those in toxin-antitoxin systems[4,5] and cell division[6,7]. Typically, the Clp system comprises the highly conserved caseinolytic protease P (ClpP) core

complex and AAA + (ATPases associated with diverse cellular activities) chaperone-unfoldases[8], including ClpX, ClpE, ClpC, and ClpA. These AAA + chaperone-unfoldases bind and use ATP hydrolysis to unfold substrate proteins, allosterically controlling the access of the unfolded, linearized substrate polypeptide through their axial pore into the proteolytic chamber of ClpP for degradation[9,10].

In many bacteria, ClpP proteases are encoded by a single *clpP* gene. For *E. coli*, *N. meningitidis*, and *H. pylori* in the absence of AAA + chaperone-unfoldases[11–14], ClpP can self-assemble into a catalytically active barrel-shaped homo-tetradecamer in an extended conformation, with each subunit comprising a characteristic Ser-His-Asp catalytic triad, positioned in a correct geometry, essential for activity[15]. For *S. aureus*, *B. subtilis*, and *F. tularensis*, ClpP proteins can be found to assemble as an inactive compressed homo-tetradecameric barrel, where the Ser-His-Asp catalytic triad is mis-aligned, rendering the complex inactive[16–21]. In the presence of activators, substrates, AAA + chaperone unfoldases, or changes in pH, the compressed, inactive ClpP homo-tetradecamer has been found to transition into the extended, active conformation[22–25]. This conformational change is characterized by the expansion of the axial pores and the rearrangement of the catalytic triad into the correct geometry, thereby restoring catalytic activity[13,22,26,27].

Although many ClpP proteases can self-assemble into catalytically active homo-tetradecameric complexes without requiring AAA + chaperone-unfoldases, they are unable to proteolyze larger protein substrates due to their constricted axial pores, which prevent larger substrates from diffusing into the catalytic chamber[28–30]. Regulated proteolysis can be facilitated by chaperone-unfoldases such as ClpX or ClpA, which remodel ClpP axial pores and facilitate the translocation of specific substrates. The binding of unfoldases to the apical surface of ClpP via their conserved LGF/IGL loops induces a conformational change in the ClpP N-terminal domain (NTD) loops that block the axial pores, thereby opening the axial pores and allowing regulated substrate entry[23,31–33].

Notably, actinobacteria encode more than one ClpP homolog. For example, *Mtb* encodes two ClpP homologs (*Mtb*ClpP1 and *Mtb*ClpP2) that are both essential for *Mtb* survival[34]. In *Mtb*, each Clp homolog has been identified to form an inactive homo-tetradecamer in vitro, exhibiting no or very weak proteolytic activity, even in the presence of chaperone-unfoldases[35,36] (*Mtb*ClpX or *Mtb*ClpC1). However, the two inactive Clp homo-tetradecamers have been found to disassemble into heptamers and reassemble into an active hetero-tetradecameric protease (*Mtb*ClpP1P2) in the presence of low-millimolar concentrations of N-blocked dipeptide activators[37,38]. Both *Mtb*ClpX and *Mtb*ClpC1 form homo-hexameric rings, which contain one or two ATP sites per subunit, respectively, and are known to interact exclusively with the *Mtb*ClpP2 ring[39]. Recent structural studies on the apo-enzyme revealed that N-blocked dipeptide activators can occupy the active sites of the *Mtb*ClpP1P2 protease while enhancing peptidase activity[38,40]. It has also been found that N-blocked dipeptide activators appear to promote the assembly of active *Mtb*ClpC1P1P2 and *Mtb*ClpXP1P2 holoenzyme complexes, leading to proteolysis of larger folded protein substrates[38]. However, the structures of mycobacterial Clp holoenzymes remain uncharacterized, leaving their mechanisms of assembly and function elusive.

In this study, we identify that bortezomib (BTZ), an anti-cancer drug inhibiting the protease activity of the human proteasome[41], exhibits anti-*Mtb* activity and is a potent activator of the *Mtb*ClpP1P2 peptidase activity[42,43]. Using activity assays and structural analysis, we show that BTZ, a boronate peptide mimetic, when sub-stoichiometrically bound to the *Mtb*ClpP1P2 active sites, can orthosterically induce a conformational change in the *Mtb*ClpP1P2 structure to activate the enzyme. We show that the BTZ-induced conformational change, which activates *Mtb*ClpP1P2, also stimulates the recruitment of AAA + chaperone-unfoldases (*Mtb*ClpX or *Mtb*ClpC1), promoting the formation of holoenzymes. We further report cryo-EM structures

of the BTZ-bound holoenzyme complexes (*Mtb*ClpC1P1P2 and *Mtb*ClpXP1P2) in various conformations, which provide insights into substrate proteolysis and its regulation in the *Mtb* Clp system.

## Results

### Sub-stoichiometric binding of *Mtb*ClpP1P2 by BTZ stimulates peptidase activity

BTZ (Fig. 1a) is known to inhibit human proteasome by binding to its protease active sites[41,44]. It has recently shown promise as a candidate anti-*Mtb* agent, presumably by targeting the essential *Mtb*ClpP1P2 complex[42]. We confirmed that BTZ exhibits a potent anti-*Mtb* activity (**MIC$_{90}$, 1.5 μM**; Supplementary Fig. 1a) and induces the formation of elongated *Mtb* cells (Supplementary Fig. 1b, c). To further test its activity on *Mtb*ClpP1P2, we co-expressed and purified the *Mtb*ClpP1P2 complex from recombinant *E. coli* (Supplementary Fig. 2a, b). Negative stain electron microscopy (EM) identified homogenous barrel-shaped tetradecameric *Mtb*ClpP1P2 complexes (Supplementary Fig. 2c). Consistent with previous reports[37,38], the complex could not cleave the small fluorescent peptide substrate PMK-AMC without an activator (Supplementary Fig. 2d). The addition of 3000 μM of the activator, N-terminally blocked dileucine (Bz-LL, Benzyloxycarbonyl-L-Leucyl-L-Leucine), resulted in moderate peptidase activity, which further increased with higher concentrations of Bz-LL (Supplementary Fig. 2d).

Interestingly, we found that as low as 5 μM of BTZ activated *Mtb*ClpP1P2 proteolytic activity toward PMK-AMC, with the activation effect of BTZ peaking at 15 μM (Fig. 1b). BTZ only completely inhibited PMK-AMC hydrolysis by *Mtb*ClpP1P2 at a concentration of 1000 μM, presumably by saturating the active sites, suggesting sub-stoichiometric binding to the active sites at lower concentrations (Fig. 1b). After confirming that BTZ can activate *Mtb*ClpP1P2 to hydrolyze small peptides, we further evaluated the effect of BTZ on the proteolysis of two larger folded protein substrates, FITC-casein and GFP-ssrA, in the presence of AAA + chaperone-unfoldases - *Mtb*ClpC1 (Fig. 1c) and *Mtb*ClpX (Fig. 1d), respectively. BTZ at concentrations of 1 and 5 μM activated the *Mtb*ClpC1P1P2 and *Mtb*ClpXP1P2 complexes, respectively, enabling efficient cleavage of FITC-casein and GFP-ssrA. Maximal activation effects were observed at BTZ concentrations of 5 μM for *Mtb*ClpC1P1P2 and 10 μM for *Mtb*ClpXP1P2 (Fig. 1c, d).

The above results demonstrate that at lower concentrations, BTZ is more potent than Bz-LL as an activator, as it activates the *Mtb* Clp system to hydrolyze small peptide or protein substrate at low micro-molar concentrations rather than millimolar concentrations. To further understand the activation of *Mtb*ClpP1P2 by BTZ, we determined the half-maximal concentration constants ($K_{1/2}$) of the hydrolysis of the peptide substrate PMK-AMC by *Mtb*ClpP1P2 at various concentrations of BTZ (Fig. 1e). A gradual decrease of substrate $K_{1/2}$ was observed in the presence of increasing BTZ concentrations, which plateaued to $K_{1/2}$ of ~1100 μM between 10 to 25 μM BTZ. Fitting the $K_{1/2}$-vs-BTZ concentration curve yielded an apparent affinity constant ($K_{app}$) of 3.7 μM (Fig. 1f), reflecting the concentration at which half of the maximum enhancement of substrate interaction occurs, as mediated by BTZ. In the same concentration range, increasing BTZ concentrations generally increased the maximal peptidase activity ($V_{max}$) of *Mtb*ClpP1P2 (Fig. 1g). The peptidase activation effect peaked at 12.5 μM BTZ. Beyond this concentration, a decrease in $V_{max}$ was observed (Fig. 1g), suggesting that BTZ above 12.5 μM starts to inhibit *Mtb*ClpP1P2 activity, likely by competing with substrate for the free proteolytic active sites (see below). Nevertheless, at 25 μM BTZ, *Mtb*ClpP1P2 still retained ~75% of the peak $V_{max}$ determined at 12.5 μM BTZ, consistent with BTZ only fully inhibiting *Mtb*ClpP1P2 at very high BTZ concentrations (Fig. 1b).

### Structure of the BTZ bound *Mtb*ClpP1P2 complex

To further understand the activation effect of BTZ, we incubated the *Mtb*ClpP1P2 complex with 15 μM BTZ, i.e., within the activating

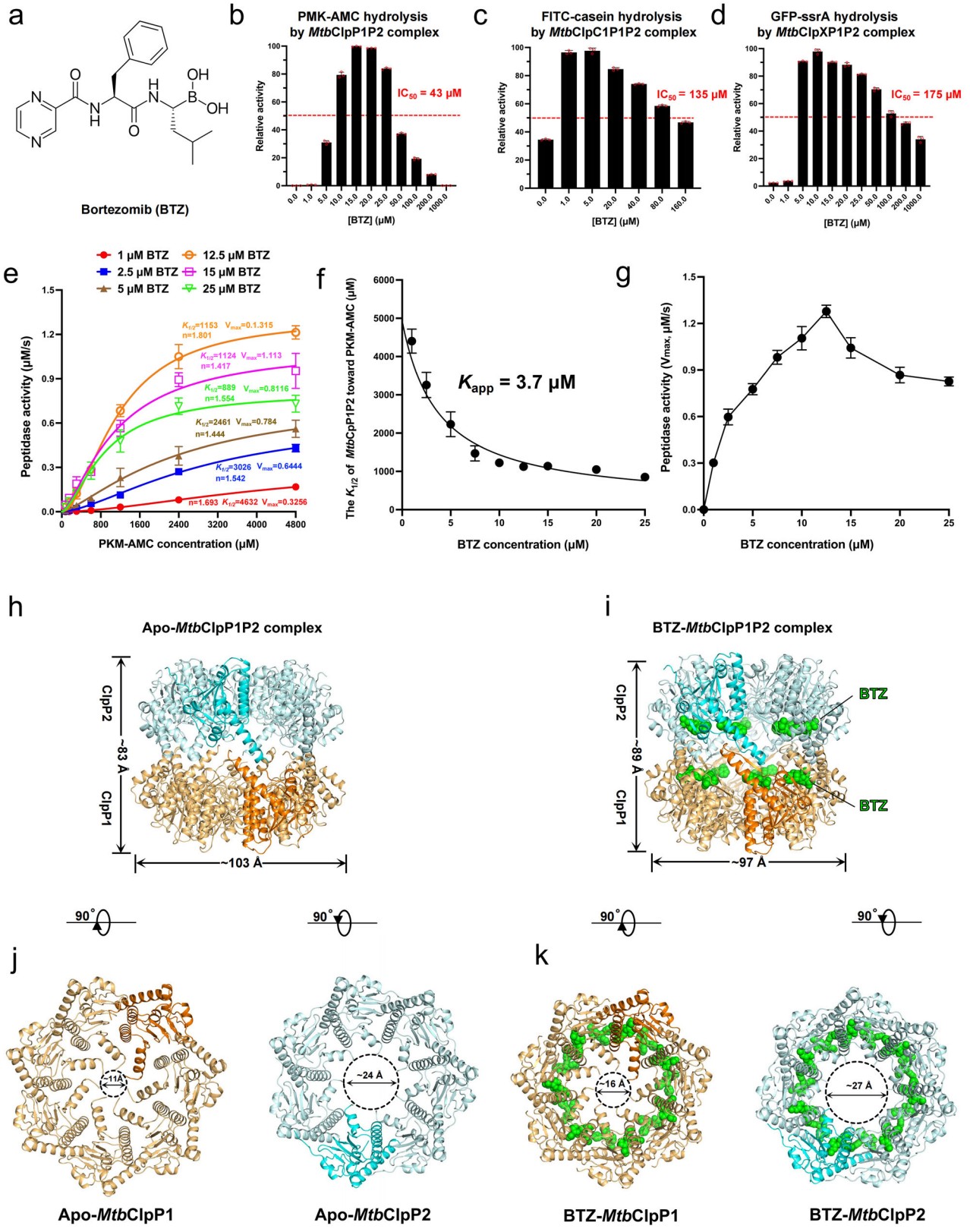

concentration range identified in the protease assay (Fig. 1b). We then determined the structure of *Mtb*ClpP1P2 bound to BTZ (BTZ-*Mtb*ClpP1P2) by cryo-EM at 2.4 Å resolution (Supplementary Fig. 3b and 4). For comparison, we also determined a 3.7 Å resolution structure of *Mtb*ClpP1P2 in the absence of BTZ (apo-*Mtb*ClpP1P2) (Supplementary Fig. 3a and 4).

In both structures, the *Mtb*ClpP1P2 hetero-tetradecamers are formed by stacking of the two *Mtb*ClpP1 and *Mtb*ClpP2 homo-heptameric rings (Fig. 1h, i). By comparison with apo-*Mtb*ClpP1P2, BTZ binding induces notable structural changes. The BTZ-*Mtb*ClpP1P2 complex adopts an "extended" conformation. The barrel of the BTZ-*Mtb*ClpP1P2 complex has a height of ~89 Å and a width of ~97 Å

**Fig. 1 | Bortezomib (BTZ) activates the *Mtb* Clp system in vitro. a** Chemical structure of bortezomib (BTZ). **b** *Mtb*ClpP1P2 complex (0.2 μM) peptidase activity was measured using the small peptide substrate PMK-AMC (200 μM) in the presence of BTZ at the indicated concentrations. **c** Peptidase activity of *Mtb*ClpC1P1P2 complex (0.2 μM) against FITC-casein (0.7 μM) in the presence of BTZ at the indicated concentrations. **d** Peptidase activity of *Mtb*ClpXP1P2 complex (0.2 μM) against GFP-ssrA (0.7 μM) in the presence of BTZ at the indicated concentrations. **b–d** In each experiment, peptidase activities were normalized to the highest activity and reported as Mean ± SD ($n = 3$); $IC_{50}$ values were estimated by fitting the inhibition part of the data to a four-parameter $IC_{50}$ equation. **e** *Mtb*ClpP1P2 (0.2 μM) peptidase saturation curves in the presence of various BTZ concentrations. Fitted enzyme kinetic parameters ($V_{max}$, $K_{1/2}$, and n) are shown. Enzyme activities are reported as Mean ± SD ($n = 3$). **f** The curve of $K_{1/2}$ versus BTZ concentrations, with the fitted $K_{app}$ value, is shown. **g** The curve of $V_{max}$ versus BTZ concentrations. **f, g** Data points in each graph are shown as mean ± SD derived from 3 independent experiments ($n = 3$). **h** Side view of the cryo-EM structure of the apo-*Mtb*ClpP1P2 complex. **i** Side view of the BTZ-*Mtb*ClpP1P2 complex structure. The *Mtb*ClpP1 heptamer (orange with one subunit highlighted in a brighter shade) and *Mtb*ClpP2 heptamer (cyan with one subunit highlighted in a brighter shade) are shown in cartoon representations. BTZ is shown as green spheres bound to the active sites of the *Mtb*ClpP1 and *Mtb*ClpP2 subunits. **j** Axial views of the apo-*Mtb*ClpP1 heptamer (left panel) and apo-*Mtb*ClpP2 heptamer (right panel). **k** Axial views of the BTZ-*Mtb*ClpP1 heptamer (left panel) and BTZ-*Mtb*ClpP2 heptamer (right panel). Source data are provided as a Source Data file.

(Fig. 1i); the widths of the ClpP1 and ClpP2 ring axial pores are ~ 16 Å and ~ 27 Å respectively (Fig. 1k). The apo-*Mtb*ClpP1P2 adopts a "compressed" conformation (~ 83 Å in height; ~ 103 Å in width) (Fig. 1h); the *Mtb*ClpP1 and *Mtb*ClpP2 ring axial pore widths are ~ 11 Å and ~ 24 Å respectively (Fig. 1j). Hence, BTZ binding induces a transition from the compressed to extended conformation and the concomitant widening of the *Mtb*ClpP1P2 axial pores. Such transition is reminiscent of the activation of *Mtb*ClpP1P2 by the dipeptide activator Bz-LL[38,40].

In the BTZ-*Mtb*ClpP1P2 structure, reconstructed imposing a C7 symmetry, BTZ densities were identified in all the 7 *Mtb*ClpP1 and 7 *Mtb*ClpP2 active sites. Weaker but substantial densities were also observed in all of the 14 *Mtb*ClpP1P2 active sites in a map reconstructed without imposing any symmetry using the same set of particles (Supplementary Fig. 3b and 4c). However, the observation that maximal *Mtb*ClpP1P2 activity occurs at ~ 15 μM BTZ (Fig. 1b and e–g), while the proteolytic activity of *Mtb*ClpP1P2 is only fully inhibited between 200 - 1000 μM BTZ (Fig. 1b), suggests that substrates can efficiently access the proteolytic active sites in the presence of 15 μM BTZ. These biochemical assay results indicate that the proteolytic active sites in *Mtb*ClpP1P2 are bound sub-stoichiometrically by BTZ in a non-saturating manner at 15 μM, the concentration used for cryo-EM sample preparation. Therefore, likely due to the cryo-EM particle alignment procedure, which is confounded by the highly symmetrical nature of the BTZ-*Mtb*ClpP1P2 complex and likely random occupancy of active sites by BTZ, the BTZ density was averaged across both occupied and unoccupied active sites. This resulted in apparent BTZ densities in all proteolytic active sites, despite biochemical data clearly indicating sub-stoichiometric BTZ binding. Nevertheless, the clear BTZ density observed (Supplementary Fig. 5a) suggests that a relatively high occupancy of *Mtb*ClpP1P2 active sites is likely achieved at 15 μM BTZ.

### Active site binding by BTZ activates *Mtb*ClpP1P2 complex

The bound BTZ molecule mimics a dipeptide with a β-strand-conformation to occupy a pocket adjacent to the catalytic triad ($S98_{P1}$, $H123_{P1}$, and $D172_{P1}$ in *Mtb*ClpP1; $S110_{P2}$, $H135_{P2}$ and $D186_{P2}$ in *Mtb*ClpP2; Fig. 2a, b).

In *Mtb*ClpP1, the amine and carbonyl groups of the BTZ backbone interact with backbone atoms of $G69_{P1}$, $I71_{P1}$, $M99_{P1}$ and $L126_{P1}$ through hydrogen bonds (Fig. 2a). The bound BTZ forms a covalent bond to the active-site serine residue ($S98_{P1}$) (Supplementary Fig. 5a), consistent with BTZ being one of the clinically approved covalent protease inhibitors[45,46]. In addition, there are also hydrophobic interactions between the bound BTZ and specific residues in the "handle" regions (α5 and β7) of *Mtb*ClpP1, namely $P125_{P1}$, $L126_{P1}$, $F143_{P1}$, $I146_{P1}$ and $M150_{P1}$ (Fig. 2a). Notably, residues $F147_{P2}$ ("X" in the *Mtb*ClpP2 "QXT" motif) and $L150_{P2}$, located at the tip of the handle region of the neighboring *Mtb*ClpP2 subunit, together with *Mtb*ClpP1 $F143_{P1}$, form hydrophobic contacts with the pyrazine ring of the *Mtb*ClpP1-bound BTZ (Fig. 2a).

In *Mtb*ClpP2, the BTZ backbone amine and carbonyl groups form hydrogen bonds with backbone atoms of $G81_{P2}$, $F83_{P2}$, $A111_{P2}$, and $S138_{P2}$ (Fig. 2b). The bound BTZ forms a covalent bond to the nucleophilic active-site serine ($S110_{P2}$) (Supplementary Fig. 5a). Hydrophobic interactions with bound BTZ mainly involve $P137_{P2}$, $L139_{P2}$, $I157_{P2}$, $M160_{P2}$, and $M164_{P2}$ in the *Mtb*ClpP2 handle regions (α5 and β6) (Fig. 2b). Notably, residues $A133_{P1}$ ("X" in the *Mtb*ClpP1 "QXT" motif) and $I136_{P1}$, located at the tip of the handle region from the neighboring *Mtb*ClpP1 subunit, are unable to contact BTZ directly, unlike their equivalents in *Mtb*ClpP2. Instead, they are part of a hydrophobic cluster with *Mtb*ClpP2 residues - $L139_{P2}$ and $I157_{P2}$, which directly contact the pyrazine ring of BTZ bound to *Mtb*ClpP2 (Fig. 2b).

The homo-heptameric *Mtb*ClpP1 and *Mtb*ClpP2 rings associate through their handle regions. Ordered handle regions have been identified as a key signature of activated ClpP proteases in other bacteria[20,25,40,47] (Supplementary Fig. 5b). In the extended conformation observed for the BTZ-*Mtb*ClpP1P2 structure, the handle regions are ordered, consistent with this being an activated conformation (Fig. 2d). In contrast, the handle regions are disordered in the apo-*Mtb*ClpP1P2 structure (res. $_{128}$GVTGS$_{132}$ in *Mtb*ClpP1 and $_{139}$LSGVIQGQF$_{147}$ in *Mtb*ClpP2) (Supplementary Fig. 5c, d). A comparison further reveals that the side chains of the BTZ-interacting residues in both the *Mtb*ClpP1 and *Mtb*ClpP2 handle regions, as well as the active sites, reorient in response to BTZ binding (Supplementary Fig. 5e, f). Due to cooperativity within the *Mtb*ClpP1P2 complex[22,40,48], we infer that sub-stoichiometric binding of BTZ also results in the reorientation of unoccupied active site residues, establishing the correct geometry among the catalytic triad residues necessary for catalysis. This explains the activation of peptidase activity (increased $V_{max}$) and enhancement of substrate interaction (decreased $K_{1/2}$) upon sub-stoichiometric BTZ binding at low BTZ concentrations (Fig. 1f, g). Such an "activation by sub-stoichiometric inhibition" mechanism has been proposed to be a general mechanism for multimeric enzymes with cooperativity[49].

Each bound BTZ molecule can form interactions bridging one *Mtb*ClpP1 and one *Mtb*ClpP2 from both rings. Such trans-interactions appear to stabilize an interaction network involving the "QXT" motifs (res. $_{132}$SAA$_{134}$ in *Mtb*ClpP1 and $_{146}$QFS$_{148}$ in *Mtb*ClpP2) located at the tip of the handle region helices, the oligomerization sensor (OS) residues[50], and the active site catalytic triad residues of *Mtb*ClpP1 and *Mtb*ClpP2. Such an interaction network expands and connects all the *Mtb*ClpP1 and *Mtb*ClpP2 subunits and presumably stabilizes the extended *Mtb*ClpP1P2 conformation associated with *Mtb*ClpP1P2 activation (Fig. 2c).

The QXT motifs have been implicated in ClpP activation[50]. Variations of QXT motif sequences among ClpPs of different bacteria have been correlated with differential ClpP activation[22,50–53]. In *Mtb*ClpP1, the QXT motif has an unconventional SAA sequence and is part of the *Mtb*ClpP2 active site (Fig. 2b, c, and e), whereas the *Mtb*ClpP2 QXT motif has a more conventional QFS sequence and is also part of the *Mtb*ClpP1 active site (Fig. 2a, c and e). In the *Mtb*ClpP1 active site, $F147_{P2}$ of the *Mtb*ClpP2 QFS sequence directly contacts the pyrazine ring of the bound BTZ (Fig. 2a), whereas the $Q146_{P2}$ participates in the inter-subunit hydrogen bond network involving QXT, OS motifs and active sites (Fig. 2c). It is noteworthy that $S132_{P1}$ in the unconventional

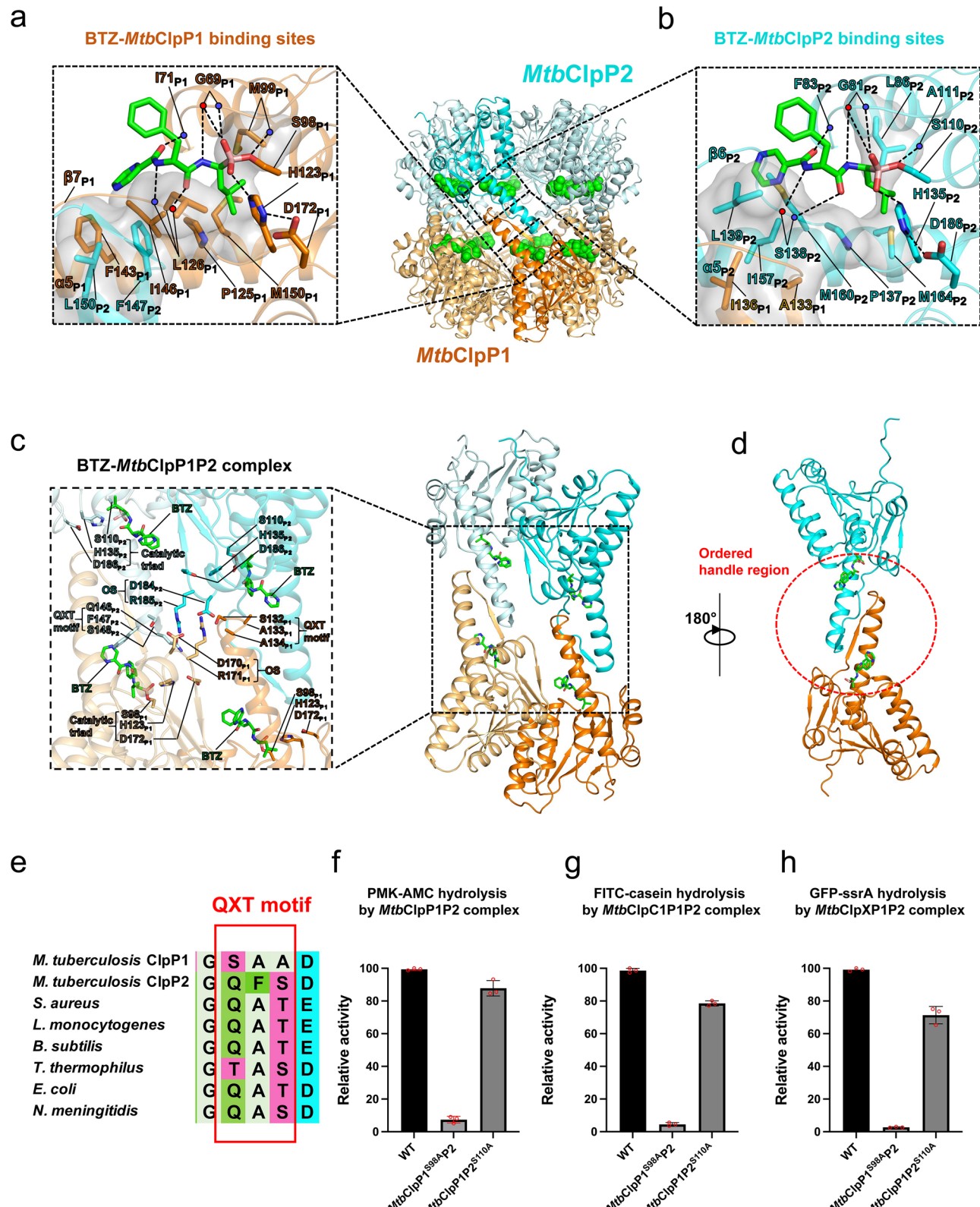

**a** BTZ-*Mtb*ClpP1 binding sites

**b** BTZ-*Mtb*ClpP2 binding sites

**c** BTZ-*Mtb*ClpP1P2 complex

**d**

**e** QXT motif

| | | QXT motif | | |
|---|---|---|---|---|
| *M. tuberculosis* ClpP1 | G | S A A | D | |
| *M. tuberculosis* ClpP2 | G | Q F S | D D | |
| *S. aureus* | G | Q A T | E E | |
| *L. monocytogenes* | G | Q A T | E E | |
| *B. subtilis* | G | Q A T | E E | |
| *T. thermophilus* | G | T A S | D D | |
| *E. coli* | G | Q A T | D D | |
| *N. meningitidis* | G | Q A S | D D | |

**f** PMK-AMC hydrolysis by *Mtb*ClpP1P2 complex

**g** FITC-casein hydrolysis by *Mtb*ClpC1P1P2 complex

**h** GFP-ssrA hydrolysis by *Mtb*ClpXP1P2 complex

*Mtb*ClpP1 QXT motif (SAA), instead of a Q, has been proposed to contribute less to the hydrogen bond network associated with *Mtb*ClpP1P2 activation[22] (Fig. 2c). In addition, compared to the *Mtb*ClpP2 equivalent F147$_{P2}$, A133$_{P1}$ within the *Mtb*ClpP1 QXT motif (SAA) loses its ability to directly contact the BTZ pyrazine ring. Consistent with previous cellular mutagenesis experiments[54], we found that mutation of the *Mtb*ClpP1 active site serine completely abolishes

*Mtb*ClpP1P2 activity, while mutation of the *Mtb*ClpP2 active site serine only has a moderate effect (Fig. 2f–h). Such results indicate that BTZ binding to *Mtb*ClpP1 active sites alone can orthosterically activate the *Mtb*ClpP1P2 complex, indicating a bias toward *Mtb*ClpP1 over *Mtb*ClpP2 in the orthosterical regulation of *Mtb*ClpP1P2. Different QXT motifs likely contribute to the functional divergence between *Mtb*ClpP1 and *Mtb*ClpP2[54].

**Fig. 2 | Structure of the BTZ bound *Mtb*ClpP1P2 complex. a, b** Detailed interactions between BTZ and active site residues in *Mtb*ClpP1 (**a**) and *Mtb*ClpP2 (**b**). Transparent molecular surfaces of selected residues involved in hydrophobic contact with bound BTZ are shown in gray. Hydrogen bonds are shown by dashed lines. **c** Cartoon representation of two *Mtb*ClpP1-ClpP2 dimers extracted from the tetradecameric BTZ-*Mtb*ClpP1P2 complex (right panel). Detailed interactions involving the QXT motif, oligomeric sensor (OS) residues, catalytic triad residues of *Mtb*ClpP1 and *Mtb*ClpP2 and BTZ are shown in the black dotted box (left panel). **d** Cartoon representation of a BTZ bound *Mtb*ClpP1-ClpP2 dimer extracted from the tetradecamer BTZ-*Mtb*ClpP1P2 complex. The ordered handle region is highlighted in the red dotted ellipse. **e** Multiple sequence alignment of the QXT motif sequences across different species of bacteria. **f** Peptidase activities of *Mtb*ClpP1P2 mutants (*Mtb*ClpP1$^{S98A}$P2 and *Mtb*ClpP1P2$^{S110A}$) (0.2 µM) against the small peptide substrate PMK-AMC (200 µM) in the presence of 15 µM BTZ. **g** Peptidase activity of *Mtb*ClpC1P1P2 mutants (0.2 µM) against FITC-casein (0.7 µM) in the presence of 10 µM BTZ. **h** Peptidase activity of *Mtb*ClpXP1P2 mutants (0.2 µM) against GFP-ssrA (0.7 µM) in the presence of 5 µM BTZ. In each experiment, peptidase activities were normalized to the WT enzyme activity and reported as Mean ± SD (*n* = 3). Source data are provided as a Source Data file.

## BTZ activation of *Mtb*ClpP1P2 facilitates recruitment of ClpC1 or ClpX

To further understand *Mtb*ClpP1P2 activation, apo-*Mtb*ClpP1P2, and BTZ-*Mtb*ClpP1P2 were compared at their *Mtb*ClpP2 apical surfaces. The comparison reveals that LGF loop binding pocket residues, L61$_{P2}$, Y75$_{P2}$, Y95$_{P2}$, V103$_{P2}$, L105$_{P2}$, L127$_{P2}$, and L204$_{P2}$, located at the apical *Mtb*ClpP2 subunit interface, move closer to each other, forming a more compact pocket (Fig. 3a, b). We further identify that the transition from the "compressed" to the activated "extended" *Mtb*ClpP1P2 conformation induced by BTZ binding also affects the spatial and relative positions of the 7 LGF loop binding pockets formed within the heptameric *Mtb*ClpP2 ring (Fig. 3c–e).

In the apo-*Mtb*ClpP1P2 complex, the LGF loop binding pockets are further apart from each other (~ 77.8 Å between distal sites, ~ 34.6 Å between proximal sites) (Fig. 3c). In the BTZ-*Mtb*ClpP1P2 complex, the LGF loop binding pockets are closer together (~ 72.6 Å between distal sites, ~ 32 Å between proximal sites) (Fig. 3d, e). We hypothesize that such changes in the spatial and relative positions of the LGF loop binding pockets likely create more suitable docking sites for the LGF loops of AAA + chaperone-unfoldases to facilitate their recruitment. The above hypothesis then may explain the results of our enzyme assays, which show that BTZ substantially stimulates the proteolytic activity of *Mtb*ClpP1P2 toward larger protein substrates in the presence of the AAA + chaperone-unfoldases *Mtb*ClpC1 or *Mtb*ClpX (Fig. 1c, d). Taken together, these structural and biochemical results hint that BTZ binding to the *Mtb*ClpP1P2 core complex induces conformational changes that could promote the recruitment of the AAA + chaperone-unfoldases.

To test the above hypothesis, we varied the concentration of *Mtb*ClpP1P2 while keeping the *Mtb*ClpC1 or *Mtb*ClpX concentration constant, in order to measure the apparent binding constants ($K_{app}$) of the *Mtb*ClpP1P2 complex toward *Mtb*ClpC1 or *Mtb*ClpX, in the presence and absence of 10 µM BTZ, a concentration shown to activate both *Mtb*ClpC1P1P2 and *Mtb*ClpXP1P2 complexes. The $K_{app}$ values for the *Mtb*ClpP1P2 complex were determined as 0.60 µM and 0.33 µM, respectively, toward *Mtb*ClpC1 and *Mtb*ClpX, in the presence of 10 µM BTZ (Fig. 3f, g). Also, under 10 µM BTZ, depending on the ClpC1:ClpP1P2 ratio, a 1.5 - 3-fold increase in peptidase activity was observed for the *Mtb*ClpC1P1P2 complex (Fig. 3f). For comparison, in the absence of BTZ, the *Mtb*ClpP1P2 complex exhibits a 2-fold weaker $K_{app}$ (1.28 µM) toward *Mtb*ClpC1, while no binding, as indicated by the lack of protease activity, was observed between *Mtb*ClpP1P2 and *Mtb*ClpX (Fig. 3f, g). These results demonstrate a moderate reduction in binding between *Mtb*ClpC1 and *Mtb*ClpP1P2 and a much more substantial abolishment of binding between *Mtb*ClpX and *Mtb*ClpP1P2, in the absence of BTZ. These results, together with those in Fig. 1c and d, highlight the effect of the orthosteric activator BTZ in promoting chaperone-unfoldase recruitment and the highly limited allosteric activation of *Mtb*ClpP1P2 by the mycobacterial chaperone-unfoldases, especially in the case of *Mtb*ClpX.

We then attempted to reconstitute the *Mtb*ClpC1P1P2 and *Mtb*ClpXP1P2 holoenzyme complexes in vitro. The purified recombinant AAA+ Walker B motif mutant chaperone-unfoldases *Mtb*ClpC1[55,56] (*Mtb*ClpC1$^{F444S-E288A-E626A}$) and *Mtb*ClpX[31] (*Mtb*ClpX$^{E187Q}$) (also see

methods) were incubated with the *Mtb*ClpP1P2 core complex in the presence of BTZ. In the incubation with *Mtb*ClpC1, the FITC-casein substrate was also added. *Mtb*ClpC1P1P2 and *Mtb*ClpXP1P2 complexes were detected by size-exclusion chromatography (SEC) (Fig. 3h, i). Cryo-EM imaging of the purified samples readily identifies the formation of the chaperone-unfoldase-engaged complexes (Supplementary Figs. 6 and 12).

## Overall structure of the *Mtb*ClpC1P1P2 complex

Three cryo-EM structures of distinct conformations were determined for the *Mtb*ClpC1P1P2 ternary complexes (Supplementary Figs. 6 and 7). These structures are referred to as "conformation 1", "conformation 2" and "conformation 3" (Fig. 4a–c).

In all three conformations, one *Mtb*ClpC1 hexamer binds one *Mtb*ClpP1P2 core complex to form a *Mtb*ClpC1P1P2 holoenzyme. Each *Mtb*ClpC1 monomer contains two AAA+ domains named D1 and D2 (Supplementary Fig. 8a). In the *Mtb*ClpC1 hexamer, domains D1 and D2 from each *Mtb*ClpC1 protomer domain-swap with D2 of the preceding protomer and D1 of the following protomer, respectively, to form a shallow spiral of two stacked pseudo-rings (D1 and D2 rings) (Supplementary Fig. 8d). The N-terminal and middle domains of *Mtb*ClpC1 (res. 1–168 and 414–477) are not visible, likely due to their flexibility. Across all three conformations, the relative height of *Mtb*ClpC1 protomers within the spiral is A (highest) > B > C > D > E (lowest), with protomer F, referred to as a "seam protomer", positioned at a height between protomers E and A (Fig. 4a–e). The lowest protomer E can also be referred to as a "lower-seam protomer". The A-F protomer interface is designated as the "seam interface[57]".

AAA + chaperone-unfoldases often bind non-specific peptides during purification[56,58–60]. Across all three conformations, a 24-residue-long peptide was observed to bind *Mtb*ClpC1. The observed cryo-EM densities allowed us to unambiguously identify this peptide as the N-terminus of FITC-casein ($_1$MKVLILACLVALALARELEELNVP$_{24}$), which was added as the substrate during sample preparation (Supplementary Fig. 8c). Similar to the reasons described for the BTZ-*Mtb*ClpP1P2 structure (above), all 14 BTZ-*Mtb*ClpC1P1P2 active sites show BTZ densities (Supplementary Fig. 8b).

## *Mtb*ClpC1P1P2 complexes show varied *Mtb*ClpC1 structures

The substrate-peptides within the three *Mtb*ClpC1P1P2 complex structures were aligned to reveal differences among the structures (Supplementary Fig. 9a). Further local alignment superposing the *Mtb*ClpP2 rings reveals that, while the *Mtb*ClpP1P2 core complexes are nearly identical across the three structures (Supplementary Fig. 9b), slight rotations are observed in the docked *Mtb*ClpC1 hexamers relative to the *Mtb*ClpP1P2 core (Supplementary Fig. 9c, d).

In all three conformations, protomers A, B, C, D, and E within the *Mtb*ClpC1 unfoldase hexamers bind the substrate-peptide via both their D1 and D2 domain pore loops. In contrast, the *Mtb*ClpC1 seam protomers do not bind the substrate-peptide (Fig. 4d, e). The nucleotide state of each *Mtb*ClpC1 ATP binding pocket has been determined by the observed ATP or ADP density and the positions of the trans-activating arginine-finger residues (R340$_{C1}$ and R341$_{C1}$ in the D1 ring; R712$_{C1}$ in the D2 ring), located in the neighboring *Mtb*ClpC1

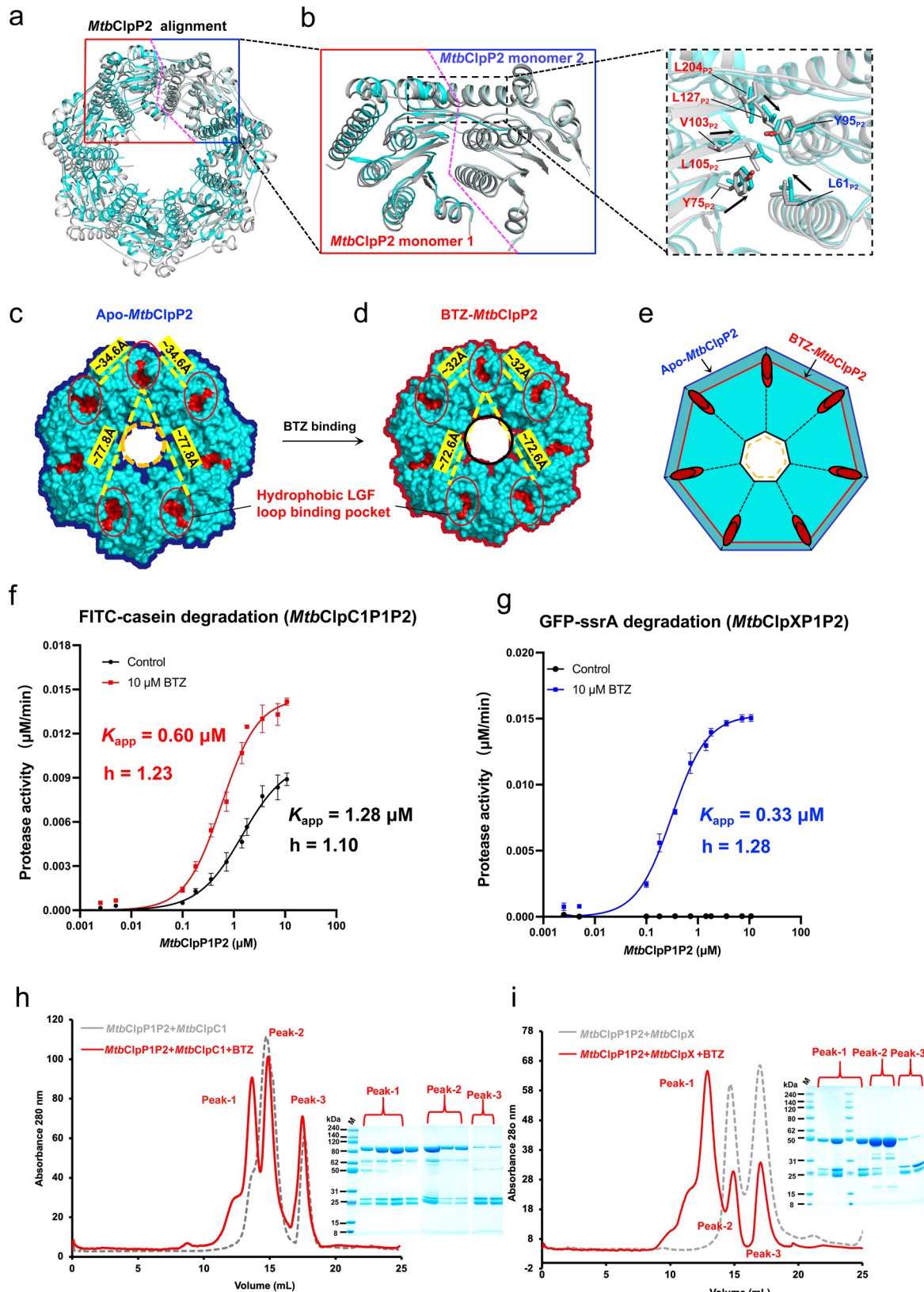

protomer[23,61] (Supplementary Fig. 10). Across the three conformations, nucleotide densities in the *Mtb*ClpC1 D1 rings are well resolved. D1 domains of protomers A-D bind ATP, whereas the lower-seam (E) and seam (F) protomer D1 domains bind ADP. More structural dynamics were observed for the *Mtb*ClpC1 D2 rings. Across the three conformations, D2 domains in the *Mtb*ClpC1 protomers A-C bind ATP. D2

nucleotide states differ somewhat in *Mtb*ClpC1 protomers D-F across the three conformations. In conformation 1, D2 of protomer D in the *Mtb*ClpC1 hexamer binds ADP; D2 of the lower-seam (E) protomer is poorly resolved with an undetermined nucleotide state, due to the protomer's high flexibility in D2; D2 of the seam (F) protomer appears to be unbound by a nucleotide. In conformation 2, D2 of protomer D

**Fig. 3 | BTZ activation of *Mtb*ClpP1P2 facilitates recruitment of ClpC1 or ClpX.**
**a** An overlap of BTZ-*Mtb*ClpP2 heptamer (cyan) and apo-*Mtb*ClpP2 heptamer (gray) based on an alignment using an *Mtb*ClpP2 dimer (red-blue box) as the reference. The red and blue half boxes mark the positions of the two *Mtb*ClpP2 monomers. **b** An overlap of the LGF loop binding pockets formed at the ClpP2 dimer interface in the apo-*Mtb*ClpP1P2 (gray) and BTZ-*Mtb*ClpP1P2 (cyan) complexes (left panel). Conserved residues in the LGF loop binding pocket are shown in the dotted black box. Residues in one *Mtb*ClpP2 subunit are labeled in red, and residues in another *Mtb*ClpP2 subunit are labeled in blue (right panel). **c, d** Surface representations of the apo-*Mtb*ClpP2 (**c**) and BTZ-*Mtb*ClpP2 (**d**) heptamers. LGF loop binding pockets are highlighted in red; the distances between pockets are indicated by the yellow dashed lines. **e** Schematic representation of the overlap between the molecular surfaces of the apo-*Mtb*ClpP2 (blue heptagon) and BTZ-*Mtb*ClpP2 (red heptagon) heptamers. Red ovals represent the LGF loop binding pockets. **f** Titration of *Mtb*ClpC1 (1.4 μM) mediated FITC-casein (0.7 μM) degradation with varied

concentrations of *Mtb*ClpP1P2 in the presence of 10 μM BTZ (red) or without BTZ (black). **g** Titration of *Mtb*ClpX (1.4 μM) mediated GFP-ssrA (0.7 μM) degradation with varied concentrations of *Mtb*ClpP1P2 in the presence of 10 μM BTZ (blue) or without BTZ (black). **f, g** Fitted apparent affinities ($K_{app}$) and hill coefficients (h) are shown. Data points in each graph are shown as mean ± SD derived from 3 independent experiments (*n* = 3). **h, i** SDS-PAGE and size-exclusion chromatography analyses of the *Mtb*ClpC1P1P2 (**h**) and *Mtb*ClpXP1P2 (**i**) complexes. Chromatogram traces in the presence of BTZ (red) and without BTZ (dashed-gray) are shown. Individual peaks are labeled, and their corresponding lanes are indicated in the SDS-PAGE gels shown in the insets. The holoenzyme peaks show the abundance of both the *Mtb*ClpP1P2 core complex and the chaperone-unfoldases, whereas the other peaks show an imbalance of the *Mtb*ClpP1P2 core complex and the chaperone-unfoldases. We repeated the experiment at least three times and obtained similar results. Source data are provided as a Source Data file.

binds ADP; D2 of the lower-seam (E) protomer appears to be unbound by a nucleotide, whereas D2 of the seam (F) protomer binds ADP. In conformation 3, D2 of protomer D binds ATP; both the lower-seam (E) and the seam (F) protomers' D2 bind ADP.

The interaction between the *Mtb*ClpC1 unfoldase hexamer and the *Mtb*ClpP1P2 core complex is mediated by the flexible LGF loops that extend from *Mtb*ClpC1 into the hydrophobic LGF loop binding pockets formed at the interface between two adjacent *Mtb*ClpP2 subunits (Fig. 5a, b). In conformations 1 and 2, the LGF loop of the *Mtb*ClpC1 lower-seam (E) protomer is unresolved, resulting in two empty LGF loop binding pockets between *Mtb*ClpP2 subunits 4 and 6 (Fig. 4f, g). In conformation 3, all 6 LGF loops are resolved and bound to *Mtb*ClpP2, with only one empty LGF loop binding pocket located between subunits 4 and 5 (Fig. 4h).

The Walker B motif mutant *Mtb*ClpC1 studied is known to have normal nucleotide binding but with greatly reduced ATPase activity[55,56]. In the presence of ATP and ADP, the latter likely generated from spontaneous ATP hydrolysis and residual ATPase activity, the three *Mtb*ClpC1P1P2 holoenzyme structures likely represent three snapshots of the holoenzyme at the beginning of the substrate translocation cycle. Nucleotide binding by the *Mtb*ClpC1 hexamers reveals nucleotide preferences at each ATPase site, offering insights into possible positions of nucleotide state change within a *Mtb*ClpC1 hexamer. All three conformations consistently show a shift from ATP to ADP binding in the *Mtb*ClpC1 D1 domain, transitioning from the D protomer to the lower-seam (E) protomer (Fig. 4d). In conformations 1 and 2, a shift from ATP to ADP binding in the *Mtb*ClpC1 D2 domain was observed upon the transition from protomer C to D (Fig. 4e, **left and middle panels**). Therefore, in both conformations, ADPs are bound in the D2 domains of protomers D, directly beneath the ADP-bound D1 domains of the lower-seam (E) protomers, due to the *Mtb*ClpC1 interprotomer D1-D2 domain-swapping (Compare Fig. 4c, e, **left and middle panels**). In conformation 3, a shift from ATP binding to ADP binding in the *Mtb*ClpC1 D2 domain was observed upon the transition from protomer D to lower-seam (E) (Fig. 4e, **right panel**), instead of C to D as observed in conformations 1 and 2. This observation suggests the possibility of a delayed ATP hydrolysis in the *Mtb*ClpC1 D2 ring. We observed that the LGF loop is structurally connected to the *Mtb*ClpC1 D2 ATPase site (Supplementary Fig. 8a). *Mtb*ClpC1P1P2 conformations 1 and 2 differ from conformation 3 in the LGF loop docking status (Figs. 4a–c, 4f–h). We speculate that the docking status of the LGF loop may affect the nucleotide state of *Mtb*ClpC1 D2, suggesting a complex coupling between ATP hydrolysis and the motion of *Mtb*ClpC1 relative to the *Mtb*ClpP1P2 core complex. Nevertheless, Across the three structures, the consistent observation of ATP binding in the top protomers, compared to ADP-bound or nucleotide-unbound ATPase sites in the seam protomers, suggests that ATP rebinding in the seam protomer is linked to its positional shift within the *Mtb*ClpC1 spiral, allowing it to become the new top protomer. (Fig. 4d, e). This, coupled

with substrate-peptide rebinding, nudges the previous top protomer, along with the others, downward within the spiral, and appears to drive substrate translocation into the *Mtb*ClpP1P2 chamber.

## Substrate interaction by *Mtb*ClpC1
The *Mtb*ClpC1 substrate channel is formed by several conserved pore loops. These include the D1 domain pore-1 loops (res. $_{260}RYR_{262}$), and D1 pore-2 loops (res. $_{294}GAGAAEGAID_{303}$), D2 domain pore-1 loops (res. $_{600}GYVG_{603}$), and pore-2 loops ($_{586}HDRFTAS_{592}$)[58] (Fig. 5a, **left panel, 5c-h and** Supplementary Fig. 11).

In the D1 ring, the D1 pore-1 loops and D1 pore-2 loops from substrate-peptide bound protomers (that is excluding the seam protomer) form 2 separate interaction networks (Fig. 5c). The conserved aromatic residues $Y261_{C1}$ of the D1 pore-1 loop act like the teeth of a gear wheel, interdigitating between alternate amino acids in the peptide substrate and forming hydrophobic contacts with its backbone[57] (Fig. 5d **and** Supplementary Fig. 11b **D1 pore-1 sequence**). Similar interactions are also made by the D2 pore-1 loop with the conserved D2 pore-1 loop residues $Y601_{C1}$ and $V602_{C1}$ (Fig. 5f, g **and** Supplementary Fig. 11b **D2 pore-1 sequence**).

The D1 pore-2 loops from the substrate-bound protomers are associated with each other through head-to-tail hydrophobic interactions (Fig. 5e). Notably, in the *Mtb*ClpC1 D2 ring, D2 pore-1 and D2 pore-2 loops from substrate-bound protomers (protomers E, F, A, B and C) also interact with each other: each D2 pore-2 loop is sandwiched by two D2 pore-1 loops via $Y601_{C1}$-$F589_{C1}$-$Y601_{C1}$ interactions (Fig. 5f and h). $F589_{C1}$ from D2 pore-2 loops, $Y601_{C1}$ and $V602_{C1}$ from D2 pore-1 loops form a hydrophobic interaction network to bind hydrophobic residues ($L6_{sub}$, $I5_{sub}$, $L4_{sub}$, and $V3_{sub}$) in the substrate-peptide (Fig. 5h). The *Mtb*ClpC1 D2 pore-2 loop residue $F589_{C1}$ is specific in mycobacteria (Supplementary Fig. 11b **D2 pore-2 loop sequences**), therefore, this interaction network, that connects the D2 pore loops, and we infer allows for their coordinated function, is likely to be specific to mycobacterial ClpC1.

## Overall structure of *Mtb*ClpXP1P2 complex
*Mtb*ClpX is another AAA + chaperone-unfoldase which can be recruited by the *Mtb*ClpP1P2 protease. *Mtb*ClpX is known to facilitate degradation of a different set of substrates from *Mtb*ClpC1 and plays a crucial role in various important cellular functions, including cell division[7], DNA replication and repair[62], as well as maintaining protein homeostasis[36,62–64].

The structure of the BTZ-*Mtb*ClpXP1P2 ternary complex was determined at a resolution of 2.3 Å (Supplementary Figs. 12 and 13). The *Mtb*ClpX hexamer is located above the *Mtb*ClpP1P2 tetradecamer complex (Fig. 6a–c). Likely due to similar reasons described for the BTZ-*Mtb*ClpP1P2 structure (above), apparent BTZ densities have been observed in all the 14 *Mtb*ClpP1P2 protease active sites (Supplementary Fig. 14a). The LGF loops from all 6 *Mtb*ClpX are bound in the respective *Mtb*ClpP2 LGF loop binding pockets. Only one LGF loop binding

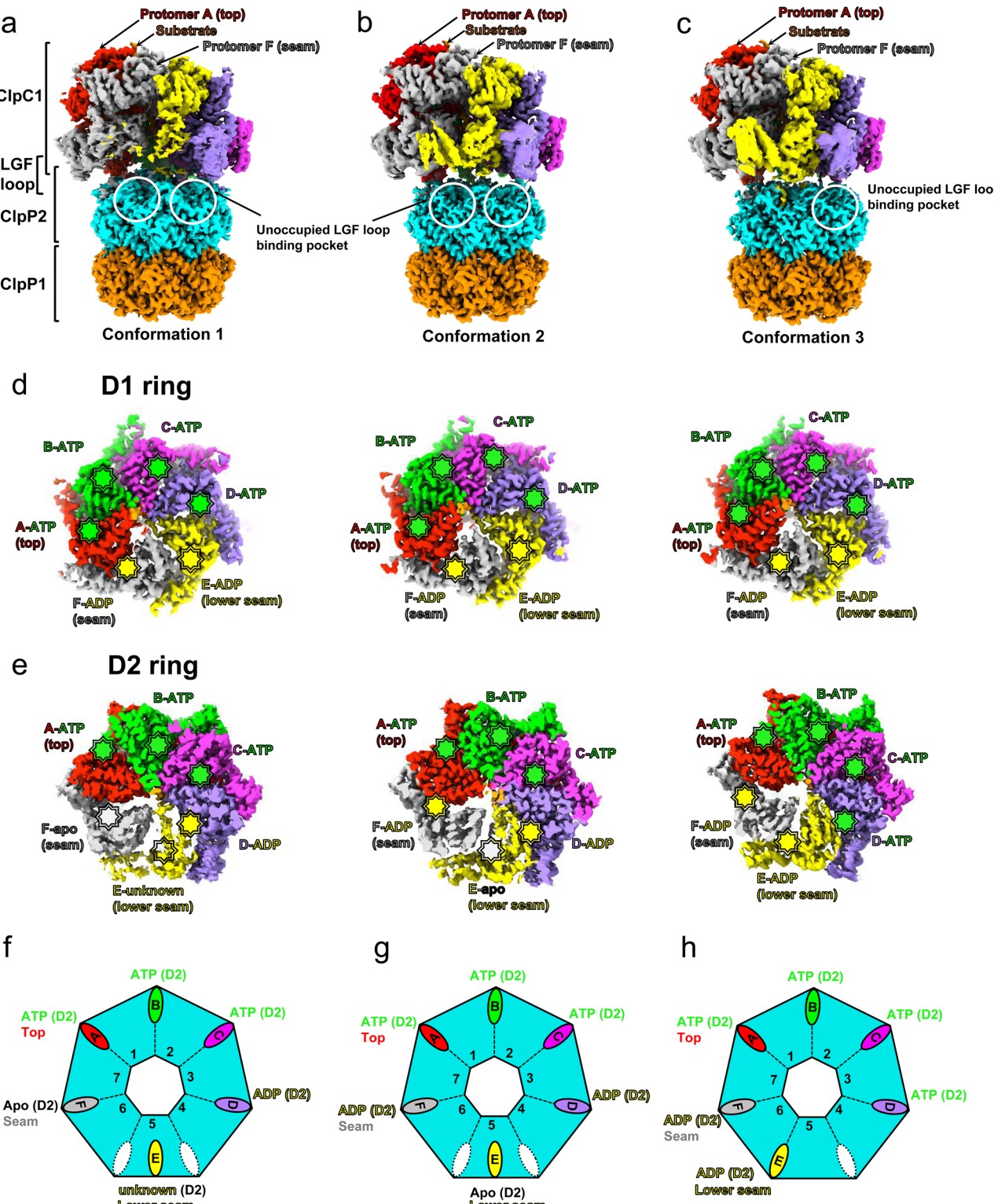

**Fig. 4 | Structural features of the *Mtb*ClpC1P1P2 complexes. a–c** Cryo-EM maps of the *Mtb*ClpC1P1P2 complexes in conformation 1 (**a**), conformation 2 (**b**), and conformation 3 (**c**) showing the *Mtb*ClpC1 hexamer bound to the *Mtb*ClpP1P2 tetradecamer. **d**, **e** Cryo-EM maps for the D1 (**d**) and D2 (**e**) rings in conformation 1 (left panels), conformation 2 (middle panels), and conformation 3 (right panels). **f**, **g** Schematic representations of the identified nucleotide state in each nucleotide binding site of the D1 (**d**) and D2 (**e**) rings in the three different conformations. ATP, ADP, apo, and unknown nucleotide states are represented by green, yellow, white, and transparent octagon stars, respectively. The nucleotide density in each site is shown in Supplementary Fig. 10. **f**–**h** Schematics showing the occupancy of *Mtb*ClpP2 LGF loop binding pockets (ovals, color-coded to show the colors of *Mtb*ClpC1 protomers of bound LGF loops; white ovals represent unbound pockets) in the three conformations.

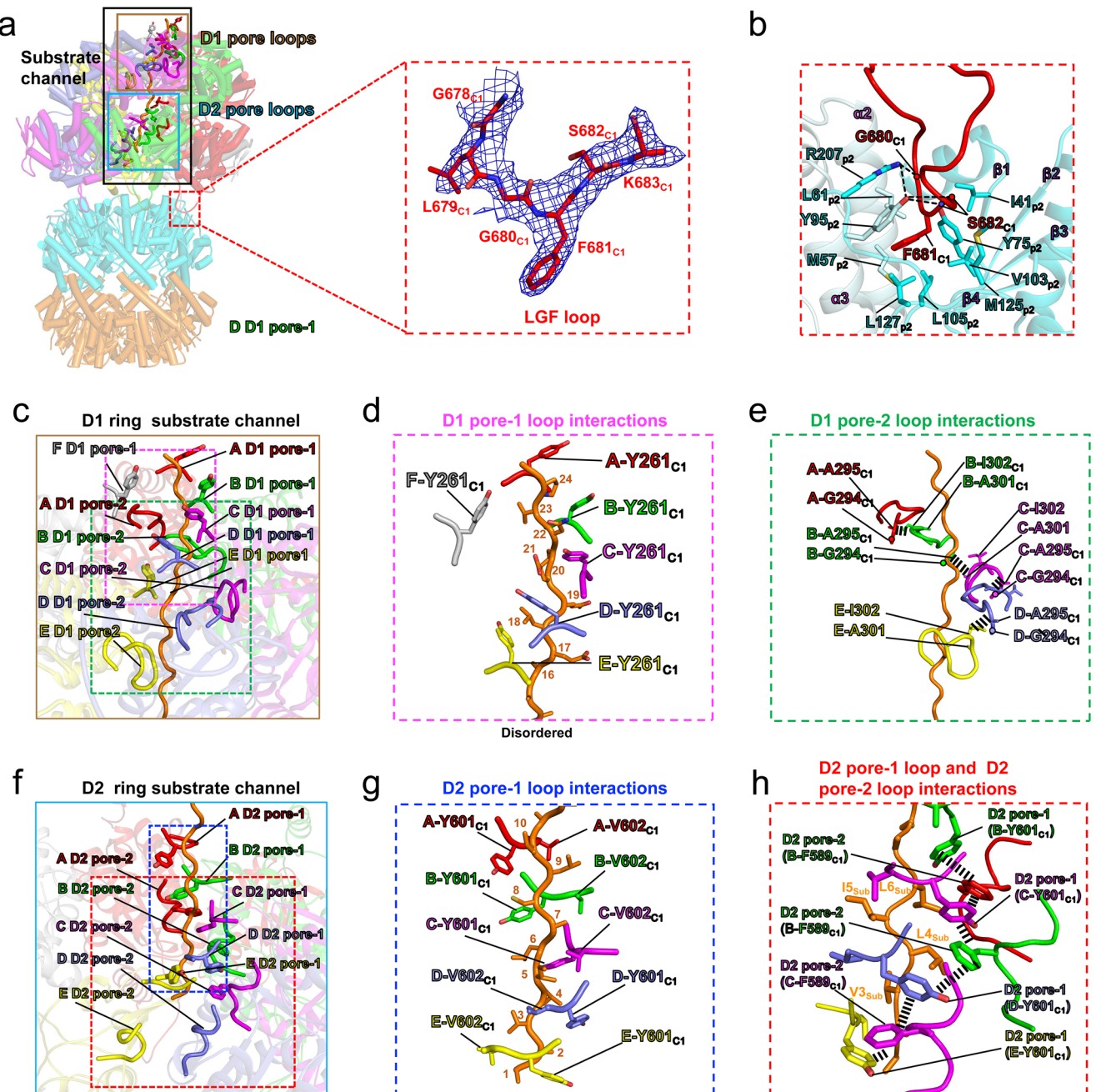

**Fig. 5 | Binding of *Mtb*ClpC1 LGF loops and structural characteristics of the substrate translocation channel.** **a** A cartoon representation showing the structure of *Mtb*ClpC1P1P2 complex in conformation 3 (left panel). Cryo-EM density (blue mesh) for the *Mtb*ClpC1 LGF loop (res. 678–683) bound to *Mtb*ClpP2 (right panel). The substrate channel in *Mtb*ClpC1 hexamer is highlighted by the black box. **b** A cartoon representation of the LGF loop (red) bound within the *Mtb*ClpP2 LGF loop binding pocket. Residues involved in hydrophobic interaction or hydrogen bonding (dashed lines) are shown and labeled. **c** A cartoon representation showing the locations of D1 pore-1 loops and D1 pore-2 loops that constitute the D1 ring substrate channel in the *Mtb*ClpC1P1P2 complex. **d** Structures of the substrate

(orange) bound D1 pore-1 loops and their Y261 residues in the *Mtb*ClpC1P1P2 complex. **e** In the *Mtb*ClpC1P1P2 complex, D1 pore-2 loop residues forming the head-to-tail hydrophobic interactions between D1 pore-2 loops are indicated by the thick dashed lines. **f** A cartoon representation showing the locations of D2 pore-1 loops and D2 pore-2 loops that constitute the D2 ring substrate channel in the *Mtb*ClpC1P1P2 complex. **g** Structures of the substrate (orange) bound D2 pore-1 loops and their Y601 and V602 residues in the *Mtb*ClpC1P1P2 complex. **h** D2 pore-1 loop, D2 pore-2 loop, and substrate residues involved in hydrophobic interactions (indicated by the thick dashed lines) are shown in the red dotted box.

pocket in the *Mtb*ClpP2 ring is left unoccupied (Supplementary Fig. 14b). Similar to the *Mtb*ClpC1P1P2 complex above, each *Mtb*ClpX LGF loop interacts with the hydrophobic LGF loop binding pocket formed between two adjacent *Mtb*ClpP2 subunits, allowing docking of the *Mtb*ClpX hexamer onto the *Mtb*ClpP2 ring (Supplementary Fig. 14c).

### *Mtb*ClpXP1P2 complex has a doubly gated substrate channel

In the *Mtb*ClpX hexamer, the substrate channel is lined with conserved pore loops, including the pore-1 loops (res. $_{154}$GYVG$_{157}$), pore-2 loops (res. $_{196}$ENPSITRDVSG$_{206}$), and RKH loops (res. $_{230}$RKH$_{232}$)[32,36] (Supplementary Fig. 14d). The *Mtb*ClpXP1P2 structure is in a substrate-free conformation, as illustrated by the blockage of the substrate channel

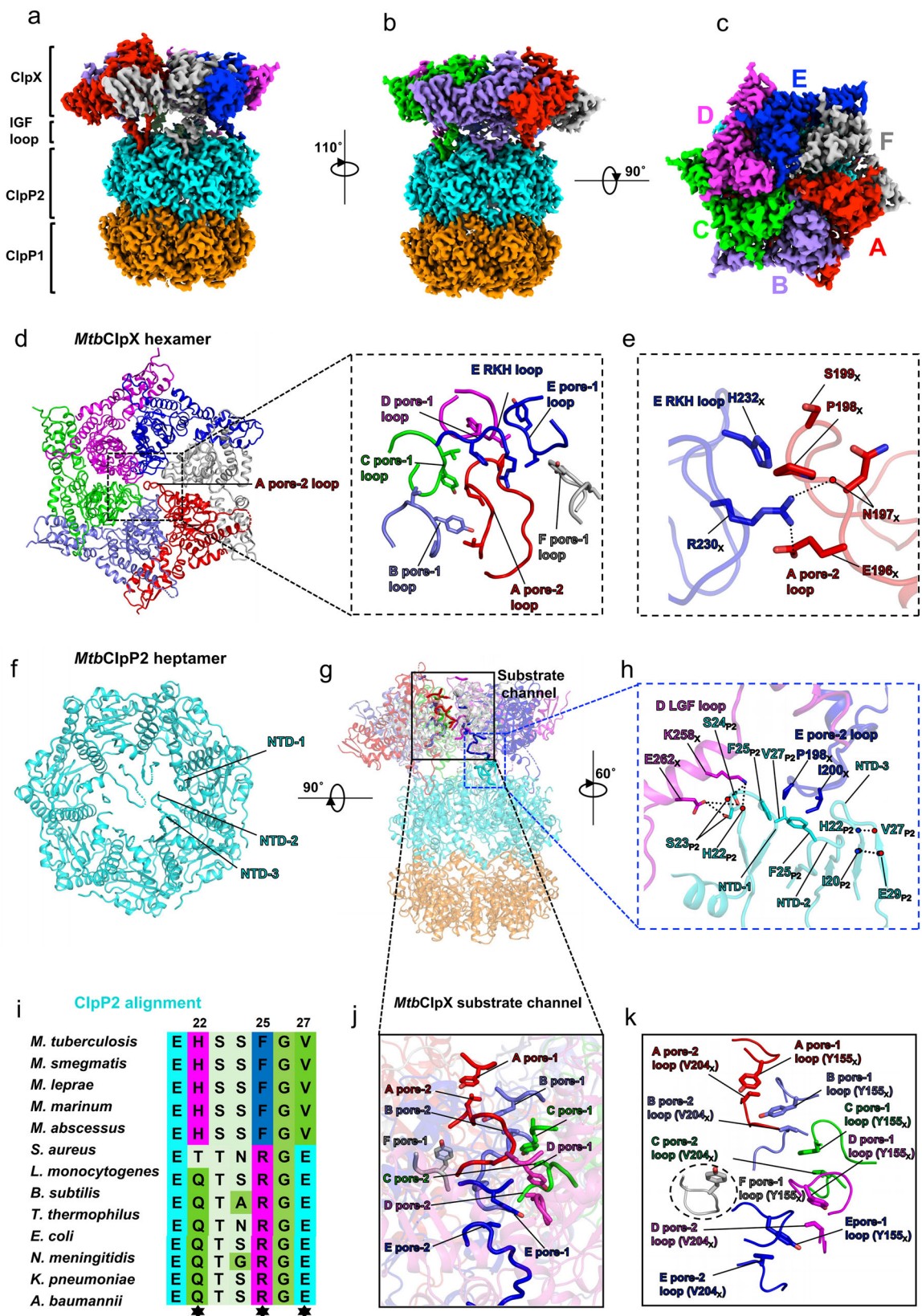

by the protomer A pore-2 loops (Fig. 6d). Specifically, the pore-2 loop of the top protomer (A) interacts with the pore-1 loops of other the protomers to form a loop cluster (Fig. 6d) that is further stabilized by the RKH loop from the bottom protomer (E) (Fig. 6e). These inter-loop contacts are mediated via hydrogen bonding and hydrophobic interactions (Fig. 6d, e).

Comparison of the $Mtb$ClpXP1P2 structure with other ClpXP structures of other bacteria[31–33] reveals that the $Mtb$ClpX hexamer sits lower in the complex (Supplementary Fig. 14e). We identify that this is due to the tilting of the $Mtb$ClpX hexamer to interact with $Mtb$ClpP2 (Fig. 6g). Specifically, residues $K258_X$ and $E262_X$ in the LGF loop of protomer D form hydrogen bonds with residues $H22_{P2}$, $S23_{P2}$ and

**Fig. 6 | Substrate translocation channel in the *Mtb*ClpXP1P2 complex. a–c**, Side (**a**, **b**), and top (**c**) views of the cryo-EM map for the *Mtb*ClpXP1P2 complex. **d** Top view of the *Mtb*ClpX hexamer in cartoon representation (left panel). The inset shows pore-1, pore-2 loops, and the RKH loop forming a blockage in the middle of the *Mtb*ClpX axial pore. Pore-1 loop residues (Y155 and V156) involved in hydrophobic interactions between pore-1 loops are shown and colored according to protomers. **e** Detailed interactions between protomer A pore-2 loop and protomer E RKH loop. Hydrogen bonds and salt bridges are shown by dashed lines. **f** A top view of the *Mtb*ClpP2 heptamer structure within the *Mtb*ClpXP1P2 complex. **g** A cartoon representation showing the location of the interactions among protomer D LGF loop, protomer E pore-2 loop, and *Mtb*ClpP2 N-terminal domains (NTDs) (blue dashed box); the *Mtb*ClpX substrate channel, formed by the conserved pore loops, is highlighted in the black box and shown in detail in panels **j** and **k**. **h** Detailed interactions involving *Mtb*ClpP2 NTD residues, protomer E pore-2 loop, and protomer D LGF loop are shown; hydrogen bonds are shown by thin dashed lines; side chains involved in hydrophobic interaction are shown in sticks. **i** Multiple sequence alignment of the ClpP2 NTD sequences. Residues involved in key interactions between *Mtb*ClpX and *Mtb*ClpP2 are highlighted with stars. **j** A cartoon representation showing the locations of pore-1 loops and pore-2 loops forming the *Mtb*ClpX substrate translocation channel in the *Mtb*ClpXP1P2 complex. **k** Detailed interactions involving pore-1 loops and pore-2 loops from each *Mtb*ClpX protomer in the *Mtb*ClpXP1P2 complex. The residues involved in hydrophobic contacts are shown and colored according to protomers.

$S24_{P2}$ located in the N-terminal domain (NTD) loop of one *Mtb*ClpP2 subunit (Fig. 6h). The pore-2 loop residues $P198_X$ and $I200_X$ of protomer E interact with residues $V27_{P2}$ and $F25_{P2}$ located in the NTD loop of an adjacent *Mtb*ClpP2 subunit via hydrophobic interactions (Fig. 6h). Due to these specific interactions, in the *Mtb*ClpXP1P2 structure, the *Mtb*ClpP2 NTD loops also block the *Mtb*ClpP2 axial pore (Fig. 6f).

In other ClpXP structures without bound substrate, such as those determined for *E. coli*[65] (*Ec*) and *Listeria monocytogenes*[31] (*Lm*), the ClpP axial pores remain open. The pore-2 loop of the lowest *Ec*ClpX protomer hardly interacts with the NTD of ClpP, while the pore-2 loops in *Lm*ClpX are partially disordered[31,65]. Furthermore, the NTD loop amino acids of *Mtb*ClpP2 that are involved in the observed interactions with *Mtb*ClpX appear, by comparison with other bacterial species, to be mycobacteria-specific (Fig. 6i). Therefore, the simultaneous closure of both the substrate channel and the *Mtb*ClpP2 axial pore appears to be a distinct feature of the *Mtb*ClpXP1P2 complex, compared to other such complexes determined in the substrate-unbound state. This feature may represent a specialized mechanism for regulating protein degradation in mycobacteria.

### Characteristics of the *Mtb*ClpXP1P2 substrate channel

The *Mtb*ClpXP1P2 pore loops possess distinct characteristics. The hydrophobic residue $Y155_X$ in the pore-1 loop interacts with the hydrophobic residue $V204_X$ in the pore-2 loop, creating in the ClpX hexamer an alternating helical arrangement of Tyr and Val residues that form a hydrophobic substrate channel (Fig. 6j, k). Similar to the interactions observed in the *Mtb*ClpC1P1P2 complex (above), these inter-loop interactions may contribute to both substrate binding and translocation.

### The nucleotide state in the *Mtb*ClpX hexamer

The cryo-EM densities indicate the presence of ATP and a magnesium ion in protomers A, B, C, and D, while protomers E and F are bound to ADP (Supplementary Fig. 14f). The trans-activating arginine-finger residues ($R370_X$) from the clockwise neighboring protomer and the sensor-II arginine ($R307_X$) residues within the same protomer interact to stabilize the bound ATP molecules (Supplementary Fig. 14i). Within the protomer E and F nucleotide binding sites, the Arg-finger residues are also displaced. Such conformational changes result in a more accessible nucleotide-binding pocket, enabling the release of the hydrolyzed ADP molecule and, presumably, subsequent ATP rebinding (Supplementary Fig. 14g, h).

## Discussion

Growing evidence indicates that the regulation of Clp systems is intricately tailored to accommodate the physiological needs and pathogenesis strategies of different bacteria[66]. Regulation of the Clp system via the Clp protease gene regulator (ClgR) has been linked to the survival of *Mtb* in macrophages[67]. Depletion of *Mtb*ClpP1 or over-expression of *Mtb*ClpP2 has been shown to be lethal for *Mtb*[34,68], again suggesting that proper regulation of the Clp system is essential for *Mtb* survival.

Therefore, understanding the regulation of the *Mtb* Clp system is essential for facilitating drug discovery targeting this system[2]. *E. coli* and *Staphylococcus aureus*, encode only one ClpP core protease subunit[13]. Their homo-tetradecamer ClpP core complexes preferentially adopt activated conformations and readily recruit AAA + chaperone-unfoldases[23,61]. In contrast, in *Thermus thermophilus*[22] and *Listeria monocytogene*[69], although their ClpP core complexes preferentially adopt inactivated conformations, docking of their cognate ClpX chaperone-unfoldases has been shown to allosterically activate the ClpP complexes. In other bacteria that encode two ClpP subunits, such as *Listeria monocytogenes*[69,70], *Pseudomonas aeruginosa*[71], and *Chlamydia trachomatis*[72,73], one of the expressed ClpP subunits (either ClpP1 or ClpP2) typically assembles into an active oligomer, and readily recruits AAA + chaperone-unfoldases. A recent study on S*treptomyces hawaiiensis* ClpP1P2 (*Sh*ClpP1P2) appears to suggest that *Sh*ClpP1P2 preferentially adopts an activated conformation and readily recruits the *Sh*ClpC1 chaperone-unfoldase[74]. Therefore, in many bacteria, Clp core proteases can be effectively regulated through chaperone-unfoldase docking and/or the resulting allosteric activation. Of note, the human mitochondrial ClpP core complex was also found to readily recruit ClpX[75], forming a ClpXP holoenzyme which was found weakly inhibited by BTZ[76].

By exploiting this regulatory feature, *Streptomyces hawaiiensis* produces a class of antibiotics, acyldepsipeptides (ADEPs), which have been shown to exert their anti-bacterial effect by deregulating the Clp system[77]. ADEPs mimic the docking of chaperone-unfoldases by binding to the LGF loop binding pockets[35,48,78]. ADEP binding has been shown to induce conformational changes in the ClpP NTD loops[48], which normally block the axial pores, resulting in the opening of axial pores in various ClpP core complexes and leading to unregulated protein degradation[35,79]. In firmicutes, nonselective degradation of essential proteins by the ADEP-activated ClpP has been attributed as the primary cause of ADEP-mediated cell death[79,80]. However, although ADEPs have been shown to bind the *Mtb*ClpP2 LGF loop binding pockets and are active against *Mtb*[80], ADEPs have been found unable to allosterically activate the *Mtb*ClpP1P2 complex without Bz-LL[40]. This suggests that ADEPs function by blocking chaperone-unfoldase docking in *Mtb* and highlighting a difference in the regulation of the *Mtb* Clp system[81]. Indeed, the *Mtb*ClpP1P2 complex has long been shown to require the presence of an activating peptide, usually an N-blocked dipeptide such as Bz-LL or Bz-LL-H (Benzyloxycarbonyl-L-Leucyl-L-Leucinal), for activity in vitro[37]. In this study, we further confirm that BTZ can function as a highly potent orthosteric activator of the *Mtb* Clp system when bound sub-stoichiometrically.

Although simultaneous ADEP and Bz-LL binding has been observed to have a synergetic effect on *Mtb*ClpP1P2 activation[35], our biochemical assays, and structural analysis reveal that BTZ binding-induced activation and the associated spatial and positional changes of the LGF loop binding pockets are crucially important for the recruitment of chaperone-unfoldases *Mtb*ClpC1 or *Mtb*ClpX. In the presence of 1 μM BTZ, *Mtb*ClpC1P1P2 complex has 3 times the activity without BTZ (Fig. 1c). In the absence of BTZ, *Mtb*ClpXP1P2 shows no activity (Fig. 1d), suggesting that *Mtb*ClpX is unable to dock to the *Mtb*ClpP1P2

core complex without prior *Mtb*ClpP1P2 activation. These observations further demonstrate that *Mtb* Clp system activation is tightly regulated, with orthosteric regulation being particularly pronounced in the *Mtb* Clp system in contrast with other bacterial species. Although BTZ has been proposed to function as a protease inhibitor[42,43], its cellular MIC$_{90}$ (1.5 μM) falls within its activating concentrations, as determined by in vitro assays. At this concentration, it remains unclear whether the intracellular BTZ concentration in *Mtb* cells reaches or exceeds the threshold required for complete inhibition of *Mtb*ClpP1P2. It remains unclear whether uncontrolled Clp system activation, potentially driven by the unregulated recruitment of unfoldases due to sub-stoichiometric BTZ binding, contributes to BTZ's anti-mycobacterial activity. In addition, BTZ has also been shown to inhibit the *Mtb* Pup-proteasome system[82–84]. Although the *Mtb* Pup-proteasome system was found to be non-essential for *Mtb* survival ex-vivo in a knock-out experiment[85,86], we cannot rule out the possibility that inhibition of the *Mtb* Pup-proteasome system contributes to BTZ's anti-mycobacterial activity.

Utilizing the activating nature of BTZ, we were able to capture the structures of the *Mtb*ClpC1P1P2 and *Mtb*ClpXP1P2 holoenzymes. The *Mtb*ClpC1P1P2 complex structures with *Mtb*ClpC1 bound to the substrate-peptide shows that 5 out of the 6 *Mtb*ClpC1 protomers bind substrate-peptide with both their D1 and D2 domains, with only the seam protomer being unbound (Supplementary Fig. 15a, e). In contrast, in the *Ec*ClpAP holoenzyme structure[23,61], also containing a chaperone-unfoldase with D1 and D2 domains, only 4 out of the 6 chaperone-unfoldase protomers bind substrate-peptide with both their D1 and D2 domains (Supplementary Fig. 15b, f). Its seam protomer is unbound, similar to *Mtb*ClpC1. However, unlike *Mtb*ClpC1, its top protomer only binds the substrate-peptide with its D2 domain, irrespective of ATP or ADP binding in its D1 domain. A similar substrate-peptide binding pattern has been observed for the recent actinomyces *Sh*ClpC1P1P2 structures, in which substrate-peptide is either weakly bound or completely unbound by the top protomer D1 domain[74] (Supplementary Fig. 15c, d, g, h).

Our *Mtb*ClpC1P1P2 structures consistently show simultaneous ATP binding in the D1 and D2 domains of the *Mtb*ClpC1 top protomers, suggesting that ATP rebinding, transition from seam to top protomer, along with the associated substrate-peptide rebinding and positional change, drives substrate-peptide translocation. Differently, ADP has been observed bound to the D1 domain of the top protomers within the chaperone-unfoldases of the *Ec*ClpAP[23] and *Sh*ClpC1P1P2[74] complexes under similar conditions. These nucleotide and substrate-peptide binding differences likely indicate variations in their mechanisms of action (Supplementary Fig. 15c, d).

In the *Mtb*ClpXP1P2 complex, we identify a *Mtb*-specific doubly gated ClpP2 substrate entry channel. This gating mechanism of the ClpP2 substrate entry channel likely represents extra regulation of substrate entry in the mycobacterial ClpXP1P2 complex by comparison with its equivalents in other bacterial species[31,65]. Recent evidence suggests that substrates of 10 residues may enter ClpP via the gap between ClpX and ClpP in *E. coli*[65]. The *Mtb*ClpXP1P2 gating mechanism that we describe may prevent such substrates from entering the *Mtb*ClpP1P2 peptidase chamber.

In summary, the reported data highlight distinct structural and regulatory features of the *Mtb* Clp system. Considering its significance for *Mtb* survival, these data should inform current anti-TB drug development efforts targeting the Clp system.

Note added in proof: while this manuscript was in revision, a preprint by Weinhäupl et al. reported a cryo-EM structure of *Mtb*ClpC1P1P2 formed in the presence of the activator Bz-LL[87].

## Methods

### Antimycobacterial activity evaluations
The autoluminescent *Mtb* H37Ra (UAlRa)[31,55] was homogenized with sterile glass beads in a 50 mL tube containing 5 mL of Middlebrook 7H9

medium plus 0.05% Tween 80, and 10% v/v oleic acid albumin dextrose catalase (OADC) supplement (7H9-OADC-Tween 80). When the Relative Light Unit (RLU) reached 2 million, various concentrations of BTZ were added to 200 μL of UAlRa broth culture (RLU diluted to 2000 ~ 4000) grown in 7H9 broth without Tween 80. DMSO served as a negative control, and rifampicin (RIF, 1 μM) was used as a positive control. RLU counts were measured daily for 12 days. The MIC$_{90}$ value was evaluated as the lowest drug concentration that can achieve an RLU$_{drug}$/RLU$_{DMSO}$ ratio of less than 10%.

UAlRa was cultured in 7H9 medium until the OD$_{600}$ reached 0.2. The experimental group was treated with 6 μM BTZ before being incubated at 37 °C with shaking at 220 rpm for 48 h. DMSO was used as a control. The samples were subsequently analyzed using a GeminiSEM 300 Ultra-High Resolution Field Emission Cryo-Scanning Electron Microscope on a 5 mm cover slide.

### Expression and purification of *Mtb*ClpP1P2
The WT *Mtb*ClpP1P2 complex was expressed and purified following a described protocol[31,55]. Briefly, *E. coli* BL21 (DE3) cells were transformed with a pETDuet-1 vector, which contains genes coding for a C-terminally Strep-II-tagged *Mtb*ClpP1 and a C-terminally 6 × His-tagged *Mtb*ClpP2. The *E. coli* cells were cultured in LB medium at 37 °C until reaching an optical density at 600 nm (OD$_{600}$) of 0.6 ~ 0.8. The cells were induced with 0.5 mM isopropylthio-beta-D-galactoside (IPTG) for ~ 12 h at 25 °C. After induction, the cells were harvested by centrifugation at 6000 × $g$ for 15 min at 25 °C. The resulting cell pellet was resuspended in Lysis Buffer 1 containing 25 mM HEPES-KOH, pH 7.5, 200 mM KCl, 10 mM MgCl$_2$, and 10% glycerol (v/v). The cell lysate was centrifuged at 20,000 × $g$ for 45 min at 25 °C before the supernatant containing the target protein was loaded onto a Ni-NTA column (Qiagen). The target protein was eluted from the column using Lysis Buffer 1 containing 200 mM imidazole. Eluted fractions containing the target protein were loaded onto a Strep-II column. The target protein was eluted with Strep Buffer containing 25 mM HEPES-KOH, pH 7.5, 150 mM KCl, 10 mM MgCl$_2$, 50 mM $d$-Desthiobiotin and 10% glycerol (v/v), before a final gel filtration using on a Superose™ 6 increase 10/300 GL column (GE Healthcare) in SEC Buffer 1 (25 mM HEPES-KOH, pH 7.5, 150 mM KCl, 10 mM MgCl$_2$).

The mutant *Mtb*ClpP1P2 complexes (*Mtb*ClpP1$^{S98A}$P2 and *Mtb*ClpP1P2$^{S110A}$) were purified using the same procedure.

### Expression and purification of *Mtb*ClpC1
*Mtb*ClpC1$^{F444S}$ mutant (F444S mutation is needed to obtain functional homogenous hexameric complex as previously described[88,89]) was cloned into the pET-28a vector, giving the target protein an N-terminal 6 × His-tag. The *E. coli* cells transformed with the expression vector were cultured in LB medium and incubated at 37 °C until the OD$_{600}$ reached 0.6 ~ 0.8. After induction with 0.5 mM IPTG for 18 h at 18 °C, the cells were pelleted before being resuspended and lysed in Lysis Buffer 2 containing 25 mM HEPES-KOH, pH 7.5, 300 mM KCl, 10 mM MgCl$_2$, 10 mM imidazole and 2 mM ATP. The lysate supernatant was loaded onto a Ni-NTA column which was washed with Lysis Buffer 2 containing 50 mM imidazole. The target protein was eluted with Lysis Buffer 2 containing 300 mM imidazole. Finally, the fractions were pooled and further purified on a Superose™ 6 increase 10/300 GL column in SEC Buffer 2 containing 25 mM HEPES-KOH, pH 7.5, 150 mM KCl, 10 mM MgCl$_2$ and 2 mM ATP.

The *Mtb*ClpC1$^{F444S-E288A-E626A}$ mutant (containing the double Walker B motif mutations, E288A and E626A, which do not affect nucleotide binding but substantially reduce ATPase activity[90,91]) was purified using the same procedure. All activity assays used purified *Mtb*ClpC1$^{F444S}$ mutant whereas all cryo-EM experiments used the *Mtb*ClpC1$^{F444S-E288A-E626A}$ mutant.

## Expression and purification of *Mtb*ClpX

WT *Mtb*ClpX was cloned into the pGEX-6P-1 vector to give the target protein an HRV 3 C Protease cleavage site-linked GST-tag. After induction (0.5 mM IPTG, 18 h at 16 °C), *Mtb*ClpX expressing *E. coli* cells were harvested by centrifugation at 6000 × *g* for 15 min at 4 °C. The pellet was resuspended and lysed in Lysis Buffer 3 containing 25 mM HEPES-KOH, pH 7.5, 200 mM KCl, 10 mM MgCl$_2$, 0.5 mM ATP, 2 mM DTT, and 10% glycerol (v/v). After centrifugation, the supernatant was loaded onto a GST column and eluted with Lysis Buffer 3 containing 50 mM reduced L-glutathione. The eluent was digested with HRV 3 C Protease at 4 °C for 26 h in Lysis Buffer 3. Finally, the fractions containing the target protein were pooled and concentrated before being purified on a Superose™ 6 increase 10/300 GL column equilibrated with SEC Buffer 3 (25 mM HEPES-KOH, pH 7.5, 150 mM KCl, 5 mM MgCl$_2$, 0.5 mM ATP and 2 mM DTT).

The *Mtb*ClpX$^{E187Q}$ mutant (containing the Walker B motif mutation E187Q, which does not affect nucleotide binding but greatly reduces ATPase activity) was purified with the same procedure.

The GFP-ssrA substrate used in the *Mtb*ClpXP1P2 protease assays, containing a C-terminal ssrA sequence (AGKEKQNLAFAA), was cloned into the pET-28a vector to give the target protein an N-terminal 6 × His-tag. The expression and purification procedures were the same as described for *Mtb*ClpC1$^{F444S}$ except that the purification buffers did not contain ATP.

All activity assays used the purified WT *Mtb*ClpX, whereas all cryo-EM experiments used the *Mtb*ClpX$^{E187Q}$ mutant.

## Enzyme assays

All assays were conducted in black 96-well plates at 25 °C. 0.4 µM WT or mutant (*Mtb*ClpP1$^{S98A}$P2 and *Mtb*ClpP1P2$^{S110A}$) *Mtb*ClpP1P2 tetradecamer complex in 50 µL of SEC Buffer 1 (25 mM HEPES-KOH, pH 7.5, 150 mM KCl, and 10 mM MgCl$_2$) was incubated with various concentrations of BTZ or Bz-LL for 30 min at 25 °C. 50 µL of SEC Buffer 1 containing 400 µM PMK-AMC substrate was added. The fluorescent signal resulting from enzyme cleavage of the substrate was monitored using a Molecular Devices FlexStation 3 plate reader with excitation at 380 nm and emission at 460 nm.

To assay FITC-casein proteolysis by the *Mtb*ClpC1P1P2 complexes in the presence of BTZ, 0.4 µM WT or mutant (*Mtb*ClpP1$^{S98A}$P2 or *Mtb*ClpP1P2$^{S110A}$) *Mtb*ClpP1P2 tetradecamer and 1.4 µM *Mtb*ClpC1$^{F444S}$ hexamer were incubated with different concentrations of BTZ in 50 µL of SEC Buffer 2 (25 mM HEPES-KOH, pH 7.5, 150 mM KCl, 10 mM MgCl$_2$ and 2 mM ATP) for 30 min at 25 °C. Subsequently, 50 µL of SEC Buffer 2 containing 1.4 µM FITC-casein was added. The increase in fluorescence (excitation at 492 nm and emission at 518 nm) was monitored at 25 °C for 1 h using a Molecular Devices FlexStation 3 plate reader. IC$_{50}$ values were estimated by fitting the inhibitory part of the activity curves to a four-parameter IC$_{50}$ equation.

To assay GFP-ssrA proteolysis by the *Mtb*ClpXCP1P2 complexes in the presence of BTZ, 0.4 µM WT or mutant (*Mtb*ClpP1$^{S98A}$P2 and *Mtb*ClpP1P2$^{S110A}$) *Mtb*ClpP1P2 tetradecamer and 1.4 µM *Mtb*ClpX hexamer were incubated with different concentrations of BTZ in 50 µL of SEC Buffer 2 for 30 min at 25 °C. Subsequently, 50 µL SEC Buffer 2 containing 1.4 µM GFP-ssrA was added. The decrease in fluorescence (excitation at 470 nm and emission at 510 nm) was monitored at 25 °C for 1 h using a Molecular Devices FlexStation 3 plate reader.

To determine enzyme kinetic parameters of *Mtb*ClpP1P2 under various BTZ concentrations, 0.4 µM of the *Mtb*ClpP1P2 tetradecamer in 50 µL of SEC Buffer 1 (25 mM HEPES-KOH, pH 7.5, 150 mM KCl, and 10 mM MgCl$_2$) was incubated with various concentrations of BTZ for 30 minutes at 25 °C. Subsequently, 50 µL of SEC Buffer 1 containing different concentrations of the PMK-AMC substrate was added. The fluorescent signal resulting from substrate cleavage by the enzyme was monitored using a Molecular Devices FlexStation 3 plate reader, with excitation at 380 nm and emission at 460 nm. The initial enzyme velocity vs PMK-AMC concentration curves were fitted to the Hill equation: $V = V_{max} \times [S]^n/(K_{1/2}^n + [S]^n)$. Where $V$ is the initial enzyme velocity, $V_{max}$ is the maximal enzyme velocity, [S] is the substrate concentration, $K_{1/2}$ is the half-maximal concentration constant, and n is the Hill coefficient. The $K_{1/2}$ versus BTZ concentration curve was fitted to a reversed binding equation: $K_{1/2} = K_{1/2max} - K_{1/2max} \times [BTZ] / (K_{app} + [BTZ]) + K_{1/2min}$, Where $K_{1/2}$ is the half-maximal concentration constant, $K_{1/2max}$ is the maximum half-maximal concentration constant, [BTZ] is the BTZ concentration, $K_{app}$ is the apparent affinity constant, and $K_{1/2min}$ is the minimum half-maximal concentration constant.

A previously published protocol[69] was followed to assay the apparent affinity ($K_{app}$) of BTZ-*Mtb*ClpP1P2 binding to *Mtb*ClpC1 or *Mtb*ClpX. 1.4 µM *Mtb*ClpC1$^{F444S}$ or *Mtb*ClpX hexamer was incubated with 5 – 21600 nM *Mtb*ClpP1P2 tetradecamer in the presence of 10 µM of BTZ in 50 µL of SEC Buffer 2 (25 mM HEPES-KOH, pH 7.5, 150 mM KCl, 10 mM MgCl$_2$ and 2 mM ATP) for 30 min at 25 °C. Subsequently, 50 µL of SEC Buffer 2 containing 2 µM FITC-casein or 2 µM GFP-ssrA was added. The change in fluorescence was monitored at 25 °C for 1 h using a Molecular Devices FlexStation 3 plate reader. Protease activity vs *Mtb*ClpP1P2 concentration was plotted and fitted to a Hill equation: $V = V_{max} \times [ClpP1P2]^h/(K_{app}^h + [ClpP1P2]^h)$. Where $V$ is the protease activity, $V_{max}$ is the maximal protease activity, [ClpP1P2] is the *Mtb*ClpP1P2 concentration, $K_{app}$ is the apparent affinity constant, and h is the Hill coefficient.

## Reconstitution of the Clp holoenzymes

0.2 µM *Mtb*ClpP1P2 tetradecamer was incubated with 5 µM BTZ for 30 min at 25 °C before 0.7 µM *Mtb*ClpC1$^{F444S-E288A-E626A}$ hexamer and 2 µM FITC-casein were added. The mixture was incubated in the presence of 2 mM ATP for 30 min at 25 °C. The mixture was purified using a Superose™ 6 increase 10/300 GL column in SEC Buffer 2 (25 mM HEPES-KOH, pH 7.5, 150 mM KCl, 10 mM MgCl$_2$, and 2 mM ATP). Fractions containing the ternary *Mtb*ClpC1P1P2 complex were identified by SDS-PAGE, pooled and concentrated for cryo-EM grid preparation.

The *Mtb*ClpXP1P2 ternary complex was prepared similarly as for *Mtb*ClpC1P1P2. 0.2 µM *Mtb*ClpP1P2 tetradecamer complex was incubated with 10 µM BTZ for 30 min at 25 °C before 0.7 µM *Mtb*ClpX$^{E187Q}$ hexamer was added. The mixture was incubated in the presence of 2 mM ATP for 30 min at 25 °C. The mixture was purified using a Superose™ 6 increase 10/300 GL column in SEC Buffer 2 (25 mM HEPES-KOH, pH 7.5, 150 mM KCl, 10 mM MgCl$_2$ and 2 mM ATP). Fractions containing the *Mtb*ClpXP1P2 ternary complex were identified by SDS-PAGE, pooled, and concentrated for cryo-EM grid preparation.

## Cryo-EM grid preparation and data collection

0.2 µM apo-*Mtb*ClpP1P2 cryo-EM protein sample was prepared in its final SEC Buffer 1. BTZ-*Mtb*ClpP1P2 cryo-EM sample was prepared by incubating 0.2 µM Apo-*Mtb*ClpP1P2 with 15 µM BTZ for 30 min at 25 °C in SEC Buffer 1. *Mtb*ClpC1P1P2 and *Mtb*ClpXP1P2 samples were in their respective final SEC Buffer 2.

Quantifoil Au R1.2/1.3 grids (300 mesh) were glow-discharged for 30 s (GloQube). 3 µL of apo-*Mtb*ClpP1P2 or BTZ-*Mtb*ClpP1P2 or *Mtb*ClpXP1P2 or *Mtb*ClpC1P1P2 complexes was applied to the glow-discharged grid at 4 °C and 100% humidity with a blot time of 2.5 s and a blot force of 4 s before flash freezing in liquid ethane in a Thermo-Fisher Vitrobot.

For the Apo-*Mtb*ClpP1P2 complex, datasets were collected using a Talos Arctica electron microscope (Thermo Fisher Scientific) with a field emission gun operating at 200 keV and equipped with a K3 Summit direct electron detector (Gatan). Micrographs were recorded at a nominal magnification of 45,000 × with a defocus range between − 0.8 and − 2.5 µm using the SerialEM software. Each movie was collected with a total dose of 60 e$^-$/Å$^2$ with a calibrated pixel size of 0.88 Å. Each movie was exposed for 1.8 s and fractionated into 27 frames.

For the BTZ-*Mtb*ClpP1P2, *Mtb*ClpXP1P2, and substrate-bound *Mtb*ClpC1P1P2 complexes. Micrographs were collected using a Titan Krios G4 microscope (Thermo Fisher Scientific) equipped with a Falcon4 direct electron detector equipped with a SelectrisX energy filter set to a 10 eV slit width. Each image was automatically collected using the EPU software (Thermo Fisher Scientific) at a nominal magnification of 165,000 ×, corresponding to a pixel size of 0.71 Å/pix. The defocus range was between − 0.8 and − 2.4 μm, with an accumulated total dose of approximately 50 e⁻/Å².

## Cryo-EM data processing

Movies were aligned with contrast transfer function (CTF) estimated in cryoSPARC Live. Particles were blob-picked and were 2D classified. For the Apo-*Mtb*ClpP1P2 complex, a total of 1,023,792 particles were selected for ab initio reconstruction, which was followed by a heterogeneous refinement. Class 6 in the ab initio reconstruction was chosen for homogeneous refinement. Topaz training was conducted, and 932,246 particles were picked and 2D classified. Finally, 712,254 particles were selected for heterogeneous refinement, and Class 1 was chosen for homogeneous refinement resulting in a 3.69 Å-resolution map.

For the BTZ-*Mtb*ClpP1P2 complex, 2,066,604 particles were picked and 2D classified. 1,266,639 particles were selected for ab initio reconstruction, followed by heterogeneous refinement. Class 5 was selected for homogeneous refinement with C7 symmetry, resulting in a final map at 2.39 Å resolution, while refinement with C1 symmetry yielded a final map at 2.56 Å resolution.

For the *Mtb*ClpC1P1P2 complex, blob picking and template picking were utilized to select particles. After 2D classification, selected particles were subjected to ab initio reconstruction. This was followed by heterogeneous refinement and homogeneous refinement, resulting in the selection of 289,374 particles for particle-picking training using Topaz. After 2D classification, 1,677,749 particles were chosen for ab initio reconstruction, and 1,133,188 particles were selected for heterogeneous refinement. Three different conformations of the complex were identified, and each obtained a map with a resolution of 2.7 Å (conformation 1), 2.53 Å (conformation 2) and 2.54 Å (conformation 3), respectively. Refined particles were subjected to particle subtraction before a local refinement with a focus on the ClpC1 hexamer and ClpP1P2 tetradecamer. Two focused maps were combined into one map in Chimera X, and each obtained a resolution map of 2.81 Å, 2.2 Å, and 2.15 Å, respectively.

For the *Mtb*ClpXP1P2 complex, particles were picked using blob picking and classified through 2D classification. A total of 3,393,352 particles were picked and classified, with 1,964,304 particles selected for ab initio reconstruction. After this, 363,029 particles were chosen for training Topaz picking and then underwent 2D classification. Following this, 1619780 particles were selected to perform ab initio reconstruction. Six classes from ab initio reconstruction were chosen for heterogeneous refinement, and Class 5 was selected for homogeneous refinement, resulting in a 2.47 Å resolution map. Refined particles were subjected to particle subtraction before a local refinement with a focus on the ClpX hexamer and ClpP1P2 tetradecamer. Two focused maps were combined into one map in Chimera X, resulting in a final resolution map of 2.24 Å.

## Model building and refinement

Model-building and refinement procedures used Chimera[92], ChimeraX[93], COOT[94], and the cryo-EM module of the Phenix package[95]. An initial model for ClpP1P2 was directly obtained from a previously determined *Mtb*ClpP1P2 structure[40] (PDB 6VGQ). The initial models for *Mtb*ClpC1 and *Mtb*ClpX were predicted using a Deep Neural Network called CroNet[96]. These initial models were then fit into the EM maps utilizing the "fit in the map" function in UCSF ChimeraX. Fitted initial models were adjusted manually in COOT[94]. Multiple rounds of real-

space refinement were performed using Phenix and Namdinator[97]. Nucleotides and BTZ were manually added using Coot, followed by real-space refinement. The structure figures were generated using PyMOL (https://pymol.org/2/), Chimera, and ChimeraX.

## Reporting summary

Further information on research design is available in the Nature Portfolio Reporting Summary linked to this article.

## Data availability

Cryo-EM maps of this study have been deposited in the Electron Microscope Data Bank (EMDB) with the accession codes EMD-39164 (Apo-*Mtb*ClpP1P2), EMD-61847 (BTZ-*Mtb*ClpP1P2, C1 symmetry), EMD-39163 (BTZ-*Mtb*ClpP1P2, C7 symmetry), EMD-39161 (BTZ-*Mtb*ClpXP1P2), EMD-39162 (BTZ-*Mtb*ClpC1P1P2, conformation 1), EMD-39157 (BTZ-*Mtb*ClpC1P1P2, conformation 2) and EMD-61842 (BTZ-*Mtb*ClpC1P1P2, conformation 3). The atomic coordinates and structure factors for the structures determined in this study have been deposited in the Protein Data Bank under the accession codes 8YD4 (Apo-*Mtb*ClpP1P2), 9JVZ (BTZ-*Mtb*ClpP1P2, C1 symmetry), 8YD2 (BTZ-*Mtb*ClpP1P2 C7 symmetry), 8YD0 (BTZ-*Mtb*ClpXP1P2), 8YD1 (BTZ-*Mtb*ClpC1P1P2, conformation 1), 8YCX (BTZ-*Mtb*ClpC1P1P2, conformation 2) and 9JVP (BTZ-*Mtb*ClpC1P1P2, conformation 3). The PDB code for the previously published structure used in this study is 6VGQ. Source data are provided as a Source Data file. Source data are provided in this paper.

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

## Acknowledgements

This study was supported by the National Key R&D Program of China (2021YFA1300903 to X.X., 2021YFA1300904 to T.Z., 2022YFC2304800 to J.Hu.), National Natural Science Foundation of China (32300152 to Y.G., 81973372 to T.Z., 32170189, 32241021 and 32361163669 to J.He., 82341085 to X.X.), Major Project of Guangzhou National Laboratory (GZNL2024A01010 to X.C.), R&D Program of Guangzhou National Laboratory (SRPG22-002 to X.X. and SRPG22-003 to J.He), China Postdoctoral Science Foundation (2022M723164 to Y.G.), Guangdong Provincial Basic and Applied Basic Research Fund (2022A1515110505 to Y.G.), Guangzhou Science and Technology Program (2023A03J0992 and 2024B03J1345 to J.Hu.), Basic Research Project of Guangzhou Institutes of Biomedicine and Health, Chinese Academy of Sciences (GIBHBRP24-02 to X.X.), Science and Technology Planning Project of Guangdong Province, China (2023B1212060050 and 2023B1212120009 to X.X. and J.He). We thank the staff at cryo-EM Facilities of GIBH-CAS, Guangzhou National Laboratory Bio-imaging Technology Platform for help with cryo-EM sample preparation and data collection. We thank Dr Petr Kuzmic of BioKin Ltd. for helpful discussion on enzyme assays. X.X. acknowledges Start-up grants from the Chinese Academy of Sciences.

## Author contributions

B.Z., Y.G., T.Z., and X.X. conceived the study; B.Z., B.L., Y.Z., and X.W. expressed and purified proteins; Y.G., C.F., H.Z., and T.Z. performed the antibacterial experiments; B.Z., Y.G., and X.H. performed enzyme activity assays with assistance from A.S.F.O., J.S., A.J.M., and S.G.B.; B.Z., H.Z., J.W., and H.Y. carried out cryo-EM data collection; H.Z., Z.L. and J.He processed cryo-EM data; H.Z., B.Z., and J.He built cryo-EM structures; B.Z. and X.X. analyzed cryo-EM structures with assistance from H.Z., J.He, A.S.F.O., J.S., A.J.M., and S.G.B.; B.Z., Y.G., H.Z., X.X., and T.Z. prepared figures; X.X., B.Z., Y.G., and T.Z. wrote the initial manuscript which was edited by all authors; X.X, T.Z., J.He, X.C., Y.G., J.Hu and N.S. acquired funding and supervised the project.

## Competing interests

The authors declare no competing interests.

## Additional information

¹State Key Laboratory of Respiratory Disease, Guangzhou Chest Hospital, Institute of Tuberculosis, Guangzhou Medical University, Guangzhou, China. ²Graduate School of Guangzhou Medical University, Guangzhou Medical University-Guangzhou Institutes of Biomedicine and Health Joint School of Life Sciences, Guangzhou Medical University, Guangzhou, China. ³Guangzhou National Laboratory, Guangzhou, China. ⁴State Key Laboratory of Respiratory Disease, Guangdong Provincial Key Laboratory of Stem Cell and Regenerative Medicine, Guangdong-Hong Kong Joint Laboratory for Stem Cell and Regenerative Medicine, Guangdong Provincial Key Laboratory of Biocomputing, Guangzhou Institutes of Biomedicine and Health, Chinese Academy of Sciences, Guangzhou, China. ⁵University of Chinese Academy of Sciences, Beijing, China. ⁶School of Basic Medical Sciences, Division of Life Science and Medicine, University of Science and Technology of China, Hefei, China. ⁷Guangxi Medical University Laboratory Animal Center, Nanning, China. ⁸Centre for Computational Chemistry, School of Chemistry, University of Bristol, Bristol, UK. ⁹School of Cellular and Molecular Medicine, Biomedical Sciences Building, University of Bristol, Bristol, UK. ¹⁰School of Biochemistry, Biomedical Sciences Building, University of Bristol, Bristol, UK. ¹¹State Key Laboratory of Virology, Wuhan Institute of Virology, Center for Biosafety Mega-Science, Chinese Academy of Sciences, Wuhan, China. ¹²Key Laboratory of Biological Targeting Diagnosis, Therapy and Rehabilitation of Guangdong Higher Education Institutes, The Fifth Affiliated Hospital of Guangzhou Medical University, Guangzhou, China. ¹³These authors contributed equally: Biao Zhou, Yamin Gao, Heyu Zhao. ✉e-mail: chen_xinwen@gzlab.ac.cn; he_jun@gibh.ac.cn; zhang_tianyu@gibh.ac.cn; xiong_xiaoli@gibh.ac.cn

