## [Transparent Peer Review file · Nature Communications]

Structural Insights into Bortezomib-Induced Activation of the Caseinolytic Chaperone-Protease System in *Mycobacterium tuberculosis*

Corresponding Author: Professor Xiaoli Xiong

Version 0:

Reviewer comments:

Reviewer #2

(Remarks to the Author)

Authors provided reasonable explanation to my concerns and I found the revisions to be a meaningful improvement.

Reviewer #3

(Remarks to the Author)

My previous criticism focused on the fact that the claim "... that bortezomib exerts its antimycobacterial activity by inducing uncontrolled activation of the Clp system" is not sufficiently supported. I pointed out that supporting this claim requires experiments with whole cells demonstrating that activation of the Clp system is the MIC driving mechanism by which bortezomib inhibits growth of mycobacteria. Such experiments have not been added and the claim remains unsupported.

Reviewer #4

(Remarks to the Author)

I appreciate the authors' efforts in revising the manuscript and addressing the reviewers' comments. The revised version addresses many of Reviewer #1's concerns. However, I have the following points to raise:

1. In the introduction, the known mechanism by which unfoldases like ClpX or ClpC activate the ClpP system through portal opening needs to be explained. All the recent cryo-EM structures of *E. coli* ClpXP or ClpAP should be cited. A discussion of the compressed and extended conformations of ClpP is currently missing. Including these details would help general readers better understand the role of unfoldases in ClpP activation.

2. The *in vivo* data (Fig. S1) do not necessarily prove that the reduced bacterial viability is due to bortezomib acting specifically on ClpP1P2. Previous studies (e.g., Maldonado et al., 2013; Rožman et al., 2020) suggest that bortezomib can also inhibit the mycobacterial Pup-proteasome. It seems more plausible that bortezomib inhibits either ClpP1P2 or the mycobacterial proteasome, rather than inducing uncontrolled activity of ClpP1P2 that leads to reduced viability. To address this, the authors must demonstrate that the *in vivo* activity of bortezomib is specific to ClpP1P2 and not an off-target effect on the Mtb proteasome, and confirm that the inhibitory concentration of bortezomib used for the *in vivo* assay is sub-stoichiometric relative to the ClpP1P2 population in the bacterium. I understand that this would require extensive additional experiments. Therefore, I suggest the authors tone down their claims and acknowledge this possibility in the manuscript

Version 1:

Reviewer comments:

Reviewer #4

(Remarks to the Author)

The authors have addressed all my comments. I do not have any further concerns.

Reviewer #2

Comment:

"Authors provided reasonable explanation to my concerns and I found the revisions to be a meaningful improvement."

Response: We thank the reviewer's insightful and constructive comments which have greatly improved our manuscript.

Reviewer #3

Comment:

"My previous criticism focused on the fact that the claim '... that bortezomib exerts its antimycobacterial activity by inducing uncontrolled activation of the Clp system' is not sufficiently supported. I pointed out that supporting this claim requires experiments with whole cells demonstrating that activation of the Clp system is the MIC driving mechanism by which bortezomib inhibits growth of mycobacteria. Such experiments have not been added and the claim remains unsupported."

Response: We thank the reviewer for this comment. We acknowledge that further experimental evidence may be needed to fully support our claim '... that bortezomib exerts its antimycobacterial activity by inducing uncontrolled activation of the Clp system' in the discussion section. Therefore, we have removed this claim and modified the sentences as the follows:

"Although BTZ has been proposed to function as a protease inhibitor, its cellular MIC₉₀ (1.5 μ M) falls within its activating concentrations, as determined by in-vitro assays. At this concentration, it remains unclear whether the intracellular BTZ concentration in Mtb cells reaches or exceeds the threshold required for complete inhibition of MtbClpP1P2. It remains unclear whether uncontrolled Clp system activation, potentially driven by the unregulated recruitment of unfoldases due to sub-stoichiometric BTZ binding, contributes to BTZ's anti-mycobacterial activity. In addition, BTZ has also been shown to inhibit the Mtb Pup-proteasome system.

Although the Mtb Pup-proteasome system was found to be non-essential for Mtb survival ex-vivo in a knock-out experiment, we cannot rule out the possibility that inhibition of the Mtb Pup-proteasome system contributes to BTZ's anti-mycobacterial activity." (Line 610-621 in the track-change MS)

Reviewer #4

Comment 1:

"In the introduction, the known mechanism by which unfoldases like ClpX or ClpC activate the ClpP system through portal opening needs to be explained. All the recent cryo-EM structures of E.coli ClpXP or ClpAP should be cited. A discussion of the compressed and extended conformations of ClpP is currently missing. Including these details would help general readers better understand the role of unfoldases in ClpP activation."

Response: We appreciate the reviewer's suggestion to provide a more detailed background. The Introduction has been revised to include an explanation of how unfoldases such as ClpX and ClpA induce activation of the ClpP system via portal opening. We have also added a discussion of the structural differences between the compressed and extended conformations of ClpP, citing recent cryo-EM studies of *E. coli* ClpXP and ClpAP. The specific modifications are as follows:

"In many bacteria, ClpP proteases are encoded by a single *clpP* gene. For *E. coli*, *N. meningitidis*, and *H. pylori* in the absence of AAA+ chaperone-unfoldases, ClpP can self-assemble into a catalytically active barrel-shaped homo-tetradecamer in an extended conformation, with each subunit comprising a characteristic Ser-His-Asp catalytic triad, positioned in a correct geometry, essential for activity. For *S. aureus*, *B. subtilis* and *F. tularensis*, ClpP proteins can be found to assemble as an inactive compressed homo-tetradecameric barrel, where the Ser-His-Asp catalytic triad is misaligned, rendering the complex inactive. In the presence of activators, substrates, AAA+ chaperone unfoldases, or changes in pH, the compressed, inactive ClpP homo-

tetradecamer has been found to transition into the extended, active conformation. This conformational change is characterized by the expansion of the axial pore and the rearrangement of the catalytic triad into the correct geometry, thereby restoring catalytic activity.

Although many ClpP proteases can self-assemble into catalytically active homo-tetradecameric complexes without requiring AAA+ chaperone-unfoldases, they are unable to proteolyze larger protein substrates due to their constricted axial pores, which prevent larger substrates from diffusing into the catalytic chamber. Regulated proteolysis can be facilitated by chaperone-unfoldases such as ClpX or ClpA, which remodel ClpP axial pores and facilitate the translocation of specific substrates. Binding of unfoldases to the apical surface of ClpP via their conserved LGF/IGL loops induces a conformational change in the ClpP N-terminal domain (NTD) loops that block the axial pores, thereby opening the axial pores and allowing regulated substrate entry.” (Line 78-101 in the track-change MS)

Comment 2:

"The in vivo data (Fig. S1) do not necessarily prove that the reduced bacterial viability is due to bortezomib acting specifically on ClpP1P2. Previous studies (e.g., Maldonado et al., 2013; Rožman et al., 2020) suggest that bortezomib can also inhibit the mycobacterial Pup-proteasome. It seems more plausible that bortezomib inhibits either ClpP1P2 or the mycobacterial proteasome, rather than inducing uncontrolled activity of ClpP1P2 that leads to reduced viability. To address this, the authors must demonstrate that the in vivo activity of bortezomib is specific to ClpP1P2 and not an off-target effect on the Mtb proteasome, and confirm that the inhibitory concentration of bortezomib used for the in vivo assay is sub-stoichiometric relative to the ClpP1P2 population in the bacterium. I understand that this would require extensive additional experiments. Therefore, I suggest the authors tone down their claims and acknowledge this possibility in the manuscript."

Response: We thank the reviewer for raising this important question. In response, we have revised the manuscript to tone down the claim that the reduction in bacterial viability was solely attributable to BTZ's specificity for ClpP1P2. The specific modifications are as follows:

“Although BTZ has been proposed to function as a protease inhibitor, its cellular MIC90 (1.5 μ M) falls within its activating concentrations, as determined by in-vitro assays. At this concentration, it remains unclear whether the intracellular BTZ concentration in Mtb cells reaches or exceeds the threshold required for complete inhibition of MtbClpP1P2. It remains unclear whether uncontrolled Clp system activation, potentially driven by the unregulated recruitment of unfoldases due to sub-stoichiometric BTZ binding, contributes to BTZ's anti-mycobacterial activity. In addition, BTZ has also been shown to inhibit the Mtb Pub-proteasome system. Although the Mtb Pup-proteasome system was found to be non-essential for Mtb survival ex-vivo in a knock-out experiment, we cannot rule out the possibility that inhibition of the Mtb Pup-proteasome system contributes to BTZ's anti-mycobacterial activity.” (Line 610-621 in the track-change MS)